# A cell-free nanobody engineering platform rapidly generates SARS-CoV-2 neutralizing nanobodies

Xun Chen [1✉], Matteo Gentili [2], Nir Hacohen[2,3,4] & Aviv Regev [1,5,6,7✉]

Antibody engineering technologies face increasing demands for speed, reliability and scale. We develop CeVICA, a cell-free nanobody engineering platform that uses ribosome display for in vitro selection of nanobodies from a library of $10^{11}$ randomized sequences. We apply CeVICA to engineer nanobodies against the Receptor Binding Domain (RBD) of SARS-CoV-2 spike protein and identify >800 binder families using a computational pipeline based on CDR-directed clustering. Among 38 experimentally-tested families, 30 are true RBD binders and 11 inhibit SARS-CoV-2 pseudotyped virus infection. Affinity maturation and multivalency engineering increase nanobody binding affinity and yield a virus neutralizer with picomolar IC50. Furthermore, the capability of CeVICA for comprehensive binder prediction allows us to validate the fitness of our nanobody library. CeVICA offers an integrated solution for rapid generation of divergent synthetic nanobodies with tunable affinities in vitro and may serve as the basis for automated and highly parallel nanobody engineering.

[1] Klarman Cell Observatory, Broad Institute of MIT and Harvard, Cambridge, MA, USA. [2] Broad Institute of MIT and Harvard, Cambridge, MA, USA. [3] Department of Medicine, Harvard Medical School, Boston, MA, USA. [4] Center for Cancer Research, Massachusetts General Hospital, Boston, MA, USA. [5] Department of Biology, Massachusetts Institute of Technology, Cambridge, MA, USA. [6] Howard Hughes Medical Institute, Chevy Chase, MD, USA. [7] Present address: Genentech, 1 DNA Way, South San Francisco, CA, USA. ✉email: xun@broadinstitute.org; aviv.regev.sc@gmail.com

Antibodies and their functional domains play key roles in research, diagnostics, and therapeutics. Antibodies are traditionally made by immunizing animals with the desired target as antigen, but such methods are time-consuming, their outcome is often unpredictable, and their use is increasingly restricted in the European Union[1]. Alternatively, antibodies can be generated and selected in vitro, where libraries of antibody-encoding DNA, either fully synthetic or derived from animals, are displayed in vitro followed by selection and recovery of those binding the intended target[2,3]. However, the adoption of such in vitro methods is still more limited than that of animal-dependent antibody generation[4], possibly due to throughput limitations and concerns over functional fitness and in vivo tolerance of antibodies generated in vitro[5]. Recent advances in antibody library design and construction, in vitro display and selection methods, post-selection binder identification and maturation have helped increase the utility of in vitro antibody generation[2]. For example, recently developed antibody library designs have been successfully used with in vitro display methods for engineering antibodies[6–8].

For typical antibodies, antigen-binding is co-determined by the variable domains of both its heavy chain (VH) and light chain (VL/VK), while camelids produce unconventional heavy-chain-only antibodies that bind to antigens solely based on the variable domain of their heavy chain, the VHH domain (also known as nanobodies). Nanobodies are increasingly used as functional antibody domains because of their small size (~14 kDa)[9] and high stability ($T_m$ up to 90 °C)[10]. Nanobody libraries have been successfully screened for binders by phage and yeast display[6,11,12]. However, the screening diversity of such cell-based systems has often been limited in practice by the efficiency of DNA library delivery into cells (e.g., the transformation efficiency of *E. coli* is typically <$10^{10}$). Conversely, cell-free approaches, such as ribosome display[13], are not limited by cell transformation and culture constraints. Despite these potential advantages, ribosome display remains underutilized compared to cell-based display systems[2], possibly due to sub-optimal efficiency and fidelity of cell-free reactions. Further optimization should open up this methodology for antibody screening and enable wider adoption of cell-free systems in antibody engineering.

Here, we develop Cell-free VHH Identification using Clustering Analysis (CeVICA), a cell-free nanobody engineering platform that integrates a synthetic nanobody library, ribosome display, and computational binder prediction using CDR-directed clustering analysis. We use CeVICA to generate more than 800 predicted nanobody binder families for the SARS-CoV-2 spike protein RBD domain. We validate the binding and virus neutralization activity of 38 predicted nanobody families and show 30 true binders and 11 neutralizers. We further apply affinity maturation and multivalency engineering to neutralizing nanobodies and generate a neutralizing agent with an IC50 of 329 picomolar. Overall, we demonstrate CeVICA's capability in rapidly discovering diverse nanobodies with good biophysical properties.

## Results

**Development of CeVICA**. To leverage the advantages of cell-free displays, we developed CeVICA (Fig. 1), an integrated platform for in vitro VHH domain antibody engineering, distinct from previous systems[7,8,14] in that it combines a design and generation method for CDR-randomized VHH/nanobody libraries, optimized ribosome display-based selection cycle with built-in background reduction, and a computational approach to perform global binder prediction from post-selection libraries. CeVICA first takes a linear DNA library as input, in which each

sequence is unique and encodes for an artificial nanobody with three fully randomized CDRs, and where the 5′ and 3′ ends of the DNA molecules contain elements required for in vitro ribosome display (Fig. 1a, see the "Methods" section). Next, CeVICA uses ribosome display to link genotype (RNAs transcribed from DNA input library that are stop codon free, and stall ribosome at the end of the transcript) and phenotype (folded nanobody protein tethered to ribosomes due to the lack of stop codon in the RNA) (Fig. 1b, see the "Methods" section). In each selection cycle (Fig. 1c, see the "Methods" section), the displaying ribosome complexes bind to an immobilized target, followed by RT-PCR of the RNA attached to the bound ribosomes, which leads to double-stranded DNA, which is then in vitro transcribed/translated in a new round of ribosome display. The double-stranded DNA in any chosen round is sequenced to obtain full-length nanobody sequences (Fig. 1d, see the "Methods" section). CeVICA then groups the sequences into clusters based on the similarity of their CDR sequences, such that each cluster represents a unique binding family (Fig. 1e, see the "Methods" section). Finally, one representative sequence from each cluster is synthesized and characterized for specific downstream applications (Fig. 1f, see the "Methods" section). The combination of linear DNA libraries (Fig. 1a), ribosome display (Fig. 1b), and selection cycles (Fig. 1c) allow the display of libraries with much larger diversity (>$10^{10}$) than methods depending on cells[15] at a similar experimental scale. As selection increases the representation of sequences encoding binders, each binder sequence leads to a cluster of sequences in the output library. Computational clustering following high throughput sequencing identifies them efficiently, promising a more comprehensive view of the landscape of binder potential.

We designed nanobody libraries containing highly random CDRs based on analysis of natural nanobody sequences and constructed the libraries using a three-stage PCR and ligation process (Fig. 1g). First, to guide our nanobody library sequence design, we analyzed the sequence characteristics of 298 unique camelid nanobodies (representing natural nanobodies) from the Protein Data Bank (PDB298) (Supplementary Data 1, see the "Methods" section), highlighting three CDR regions, CDR1–3[9], separated by four regions of low diversity, frame1–4 (analysis of a larger dataset containing 1030 sequences from abYsis showed the same sequence features. Supplementary Fig. 1a, Supplementary Data 2). The four frames share high homology with human IGHV3-23 or IGHJ4 (Supplementary Fig. 2a, b), and most of the remaining non-identical residues are present in other human IGHV genes (Supplementary Fig. 2c). We used consensus sequences extracted from this profile to design nanobody DNA templates encoding the four frames (Fig. 1g) and included additional frames to the final mixture of frame templates (Supplementary Data 3, see the "Methods" section), based on well-characterized nanobodies[10,16]. The mixture of nanobody frames serves as templates in PCR reactions, where DNA oligonucleotides with a 5′NNB sequence were used to introduce randomization in CDRs, while hairpin DNA oligonucleotides were used to block ligation of one end of the PCR product (Fig. 1g and Supplementary Fig. 3, see the "Methods" section). We introduced 7 random amino acids for CDR1, 5 for CDR2, and 6, 9, 10, or 13 for CDR3 to match the most commonly observed CDR lengths in natural nanobodies. CDR3s longer than 13 amino acids only account for a minority of natural nanobodies (36%, Supplementary Fig. 1a, Supplementary Data 2) and were not included in our nanobody library. CDRs randomized in earlier stages are subject to duplication in later stages that reduce their diversity. We thus chose to randomize CDR2 first, followed by CDR1, and then CDR3, imposing a diversity hierarchy of CDR3 > CDR1 > CDR2, because this is the overall ranking of

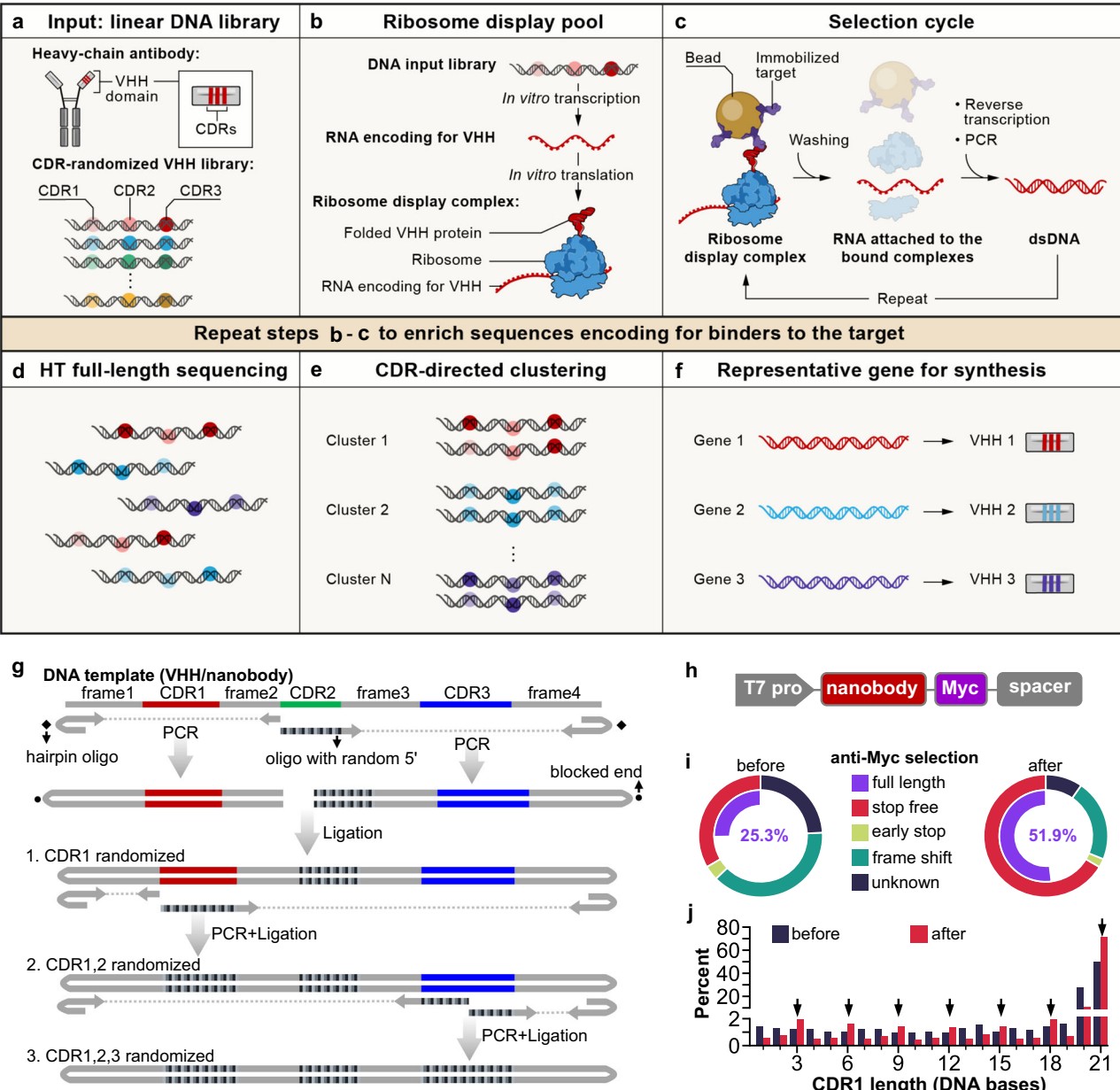

**Fig. 1 A cell-free nanobody engineering platform for rapid isolation of nanobodies from large synthetic libraries. a** The workflow takes linear DNA library as input. **b** Ribosome display links genotype (RNAs transcribed from DNA input library that are stop codon free, and stall ribosome at the end of the transcript) and phenotype (folded VHH protein tethered to ribosomes due to the lack of stop codon in the RNA). **c** Selection cycle that enriches DNA encoding for nanobodies that binds immobilized targets. **d** High throughput sequencing of full-length VHHs. **e** Sequences are grouped into clusters based on the similarity of their CDRs, each cluster is distinct and represents a unique binding family. **f** The system outputs one representative sequence from each cluster to be synthesized and characterized for specific downstream applications. **g** Workflow for generating VHH/nanobody library. CDR randomization was introduced by PCR using a hairpin oligo (blocks DNA end from ligation) and an oligo with a random 5' sequence, followed by orientation-controlled ligation. Three successive PCR plus ligation steps randomize all three CDRs. **h** The final DNA library sequence structure. **i** One round of ribosome display and anti-Myc selection was performed after randomization of CDR1 and CDR2. The pie chart shows the percentage of indicated sequence categories before and after anti-Myc selection. **j** Length distribution of DNA region encoding CDR1 of the nanobody library before and after anti-Myc selection. Arrows indicate all correct-frame lengths showing an increased percentage after anti-Myc selection. Source data for **i** and **j** are provided in the Source Data file.

diversity we observed in CDRs in natural nanobodies (Supplementary Fig. 1a, c). The sequence profile of the resulting randomized nanobody library met our design objectives, and largely mirrored the sequence features of natural nanobodies (Supplementary Fig. 1 and Supplementary Data 2). Notably, our library design differs from previous synthetic nanobody library designs[6–8] in several key ways: we defined CDR boundaries and

length differently (based on our analysis of natural nanobodies, Supplementary Data 2, see the "Methods" section), for example, in CDR2 (Supplementary Fig. 4), and we performed complete randomization of all CDR positions with NNB codons (and do not avoid, for example, cysteines in these positions) to maximize amino acid sequence possibilities. Finally, the nanobody DNA library contains an upstream T7 promoter to allow transcription

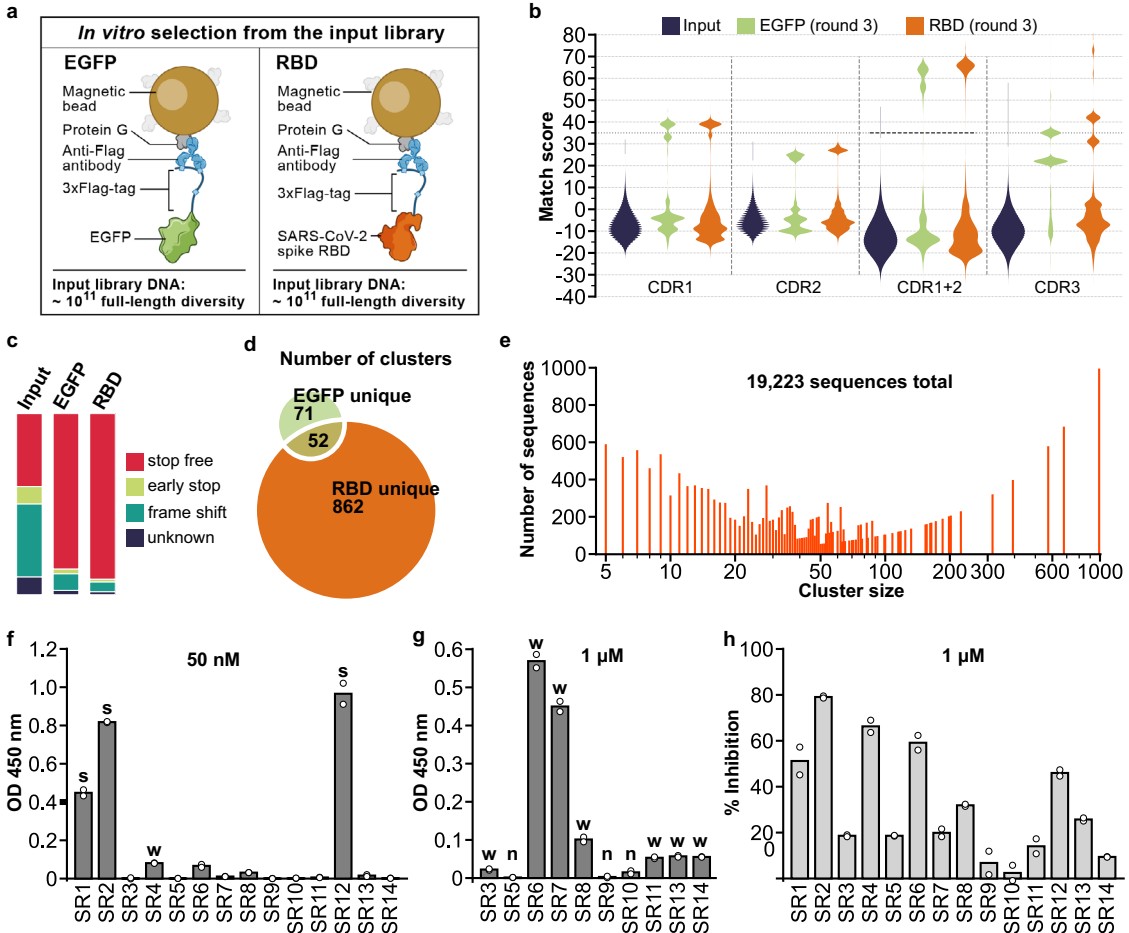

**Fig. 2 Isolation and characterization of synthetic nanobodies that bind SARS-CoV-2 spike RBD. a** Immobilization strategy for the target proteins: 3×Flag-tagged EGFP or RBD. **b** Pair-wise CDR match scores (based on BLOSUM62 matrix) were calculated for 2000 randomly selected sequences from input library and output libraries after 3 rounds of selection. High match score populations appeared in the output libraries. Combining CDR1 and 2 match scores further separated high and low score populations and a match score of 35 (black dashed line) was chosen as the cut-off for downstream clustering analysis. **c** Percentage of indicated sequence categories in the input library and output libraries (EGFP, RBD). **d** Number of unique and shared clusters identified in EGFP and RBD output libraries. **e** Number of sequences for each size of RBD unique clusters. **f** ELISA assay revealed 3 strong binders ("s") to RBD, 8 weak binders ("w"), and **g** 3 non-binders ("n", background-subtracted OD 450 nm <0.02) among the 14 nanobodies chosen for characterization. **h** SARS-CoV-2 S pseudotyped lentivirus neutralization assay showed 6 nanobodies inhibiting infection >30% at 1 μM on HEK293T-expressing ACE2 and TMPRSS2. Data shown are two technical replicates, bar height: mean, circle: the value of each replicate. Source data for **c**, **e**, **f**, **g**, and **h** are provided in the Source Data file.

of nanobody RNA, a 3×Myc tag, and a spacer downstream of the nanobody coding region that stalls peptide release, to enable ribosome display (Fig. 1h).

To test the performance of our library in ribosome display, and to reduce unproductive sequences, such as nanobodies that contain frameshifts or early stops, we ribosome displayed a library only with randomized CDR1 and CDR2 and performed one round of anti-Myc selection. Functional nanobody sequences will express the Myc tag at the C-terminal of nanobody and are expected to be enriched after anti-Myc selection. Indeed, there was a large decrease of unproductive sequences and an increase of full-length nanobodies (from 25.3% to 51.9%) after anti-Myc enrichment (Fig. 1i, see the "Methods" section). At the DNA level, there was an increase of all in-frame CDR1 DNA lengths and a decrease of frame-shift lengths (Fig. 1j). We used the resulting full-length enriched CDR1 and 2 randomized libraries as the PCR template for randomization of CDR3. The final library with all three CDRs randomized (hereafter, "the input library") contained 27.5% full-length sequences, and $3.68 \times 10^{11}$ full-length diversity per μg of library DNA.

**Binder selection for RBD and EGFP**. We performed in vitro selection from the input library for sequences that encode binders to two target proteins: EGFP and the receptor-binding domain (RBD) of the spike protein of SARS-CoV-2[17] (Fig. 2). We fused each of the two proteins with a 3×Flag tag and immobilized them on beads coated with protein G and anti-Flag antibody (Fig. 2a). For each screen, we used input library DNA corresponding to ~1 × 10^11 full-length diversity and performed 3 rounds of selection. After round 3, with an optimized PCR approach that minimized loop shuffling[18] (see the "Methods" section), RNA yield markedly increased in both screens (Supplementary Fig. 5a) and the recovered sequences were primarily composed of *E. coli* ribosomal RNAs and nanobody library RNA (e.g., Supplementary Fig. 5b). Comparing the input and output library sequences shows a 2.3-fold increase in the proportion of stop-free nanobody sequences after 3 rounds of selection (Fig. 2c), fitting our expectation that successful binding to targets depends on intact nanobody structure.

We identified target-specific binders by computationally clustering CDR sequences enriched after selection into families

while accounting for sequencing errors (see the "Methods" section). First, we examined the distribution of the sequence match scores (see the "Methods" section) of CDRs between randomly selected pairs of sequences within a library, and compared these distributions for each CDR between the input and output libraries (Fig. 2b, see the "Methods" section). In the pre-selection input libraries, the mean match score is low and the distribution is unimodal, as expected given the randomization; whereas after selection, there is a multi-modal distribution, with one low mode (similar to the input) and at least one high mode (Fig. 2b), which is further distinguished when combining the CDR1 and CDR2 match scores (Fig. 2b). This high mode should reflect binders enriched by the selection rounds. Notably, sequences with a high match score in one CDR are more likely to have a high match score in other CDRs (Supplementary Fig. 5c–f). We clustered the likely binder sequences exceeding a combined (two CDRs) match score threshold (Fig. 2b, dashed horizontal line), yielding 862 unique clusters for RBD and 71 for EGFP, and 52 clusters shared by the two targets (Fig. 2d, Supplementary Data 4 and 5). Notably, RBD unique clusters span a wide range of cluster sizes (Fig. 2e). Conversely, the shared clusters represent background binders and are excluded from further analysis, because they do not show specific binding to either EGFP or RBD.

Focusing on RBD binders, we chose one representative nanobody gene from each of the 14 top-ranking (ranked by cluster size) RBD unique clusters and validated it for spike RBD binding and SARS-CoV-2 pseudovirus neutralization (Fig. 2f–h, see the "Methods" section). RBD binding ELISA assays of the 14 tested nanobodies (SR1–14) showed three strong binders (SR1, 2, 12), 8 weak binders (SR3, 4, 6, 7, 8, 11, 13, 14), and 3 non-binders (Fig. 2f, g). SARS-CoV-2 S pseudotyped lentivirus neutralization assays revealed 6 nanobodies inhibiting infection above 30% at 1 μM (Fig. 2h), which included the 3 strong binders and three of the weak binders (SR4, 6, 8).

**Validation of NNB codon fitness for binder selection**. We next compared input, output, and natural CDR sequence distributions to assess whether starting with a fully random CDR amino acid profile could be generally detrimental to the fitness of binders and whether selection output mimics a natural amino acid distribution. In particular, in natural nanobodies (PDB298, Supplementary Data 2), CDR1 and CDR2 are less diverse than CDR3 with an amino acid profile that favors certain residues (Supplementary Fig. 1a, c), and previous synthetic nanobody library designs sought to recapitulate the CDR1 and CDR2 amino acid preferences of natural nanobodies[6–8]. Conversely, we used fully randomizing NNB codons to encode all CDR positions. In principle, such a design might not be ideal if the natural CDR1 and CDR2 amino acid profiles are required for functional nanobodies; alternatively, it may allow us to recover possibilities not captured by libraries pre-biased by natural sequence distributions.

To determine whether our fully random CDR amino acid profile is detrimental to the fitness of binders, we compared the CDR amino acid profile of 932 representative sequences across all unique clusters from both the EGFP and RBD output libraries ("output binders") (Supplementary Fig. 6, Supplementary Data 2) to the sequence profiles of either the input library or natural nanobodies (Supplementary Fig. 1a, b). We reasoned that if the amino acid profile in the input library leads to a distribution of proteins that are less fit in binding, the binder selection process should shift this distribution to a more fit profile in the output library, such that there is a low correlation between the amino acid profiles of the input library and output binders. Surprisingly,

there was an overall smaller shift in CDR1 and CDR2 compared to CDR3, as indicated by higher Spearman correlation coefficients (Fig. 3a–c, mean Spearman correlation = 0.73, 0.73, and 0.64 respectively), and shorter distances (as the RMSE relative to $y = x$ line, see the "Methods" section, Fig. 5d, e, mean RMSE = 2.96, 2.40 and 3.51, respectively), implying that a fully random profile at CDR1 and CDR2 may not have had a substantial binding fitness cost at most positions, whereas CDR3 not only shifted away from the input profile, it was even further shifted from the natural profile (Fig. 3d, e). Moreover, the correlation of amino acid profiles between output binders and natural nanobodies is significantly less than between output binders and input library at most CDR positions (Fig. 3). A few positions (CDR1 position 7 and CDR3 position 1–3) had much lower input–output binders Spearman correlation coefficients and higher RMSE than most positions. This suggests that these positions may benefit from specifically designed amino acid profiles (to adjust off-diagonal amino acids frequencies (Fig. 3b) to fit the diagonal line), even though their input distributions were not particularly distinct from the natural sequence distribution compared to other positions (Fig. 3a, d). We observed similar results when we used a larger collection of 1030 natural nanobody collections from abYsis (www.abysis.org, abYsis1030) to calculate the natural profile (Supplementary Fig. 7). Thus, the output binder CDR profile is predominantly influenced by the input library rather than by selection towards a natural nanobody profile, a natural nanobody CDR amino acid profile is not required for nanobody binding, and a fully random CDR design offers high diversity without a major binding fitness cost (although may have other fitness drawbacks in vivo).

**Affinity maturation effectively improves nanobody function**. To perform affinity maturation, a critical stage in antibody development in animals, we designed and performed an affinity maturation strategy based on CeVICA to increase the affinity of RBD-binding nanobodies (Fig. 4a, see the "Methods" section). We used error-prone PCR to introduce random mutations across the full-length sequence of six selected nanobodies (SR1, 2, 4, 6, 8, 12) and generated the mutagenized library. We used a library size of $4.18 \times 10^{10}$ (sufficient to cover the full diversity of nanobodies with three mutations per sequence) as input and performed three rounds of stringent selection. We sequenced the libraries pre- and post-affinity maturation and observed about 3 mutations per sequence in the pre-library and about 2 mutations per sequence in the post-library (Fig. 4a). We calculated their position-wise amino acid profiles, and determined, for each nanobody, the change in each amino acid proportion at each position, generating a percent point change table. We defined putative beneficial mutations as those with a percentage point increase above a set threshold (Fig. 4b, see the "Methods" section and Supplementary Data 6), highlighting between 8 and 25 putative beneficial mutations for each of the selected nanobodies. Finally, we assembled a list of identified putative beneficial mutations for each nanobody and incorporated different combinations of them into each nanobody parental sequence to generate multiple mutated variants of each nanobody for final assessment (Supplementary Data 7).

Variants in the SR4 and SR6 families had both increased binding and neutralization, while the SR2 and SR12 family variants had only increased neutralization but not increased binding, based on ELISA-binding assays and pseudotyped virus neutralization assays (Fig. 4c, d). Multiple nanobody variants outperformed VHH72, a previously described nanobody that neutralizes SARS-CoV-2 pseudoviruses[19], in binding (e.g., SR12c3), neutralization (e.g., SR4t6), or both (e.g., SR6c3) (Fig. 4c, d and Supplementary Data 8). Neutralization and binding

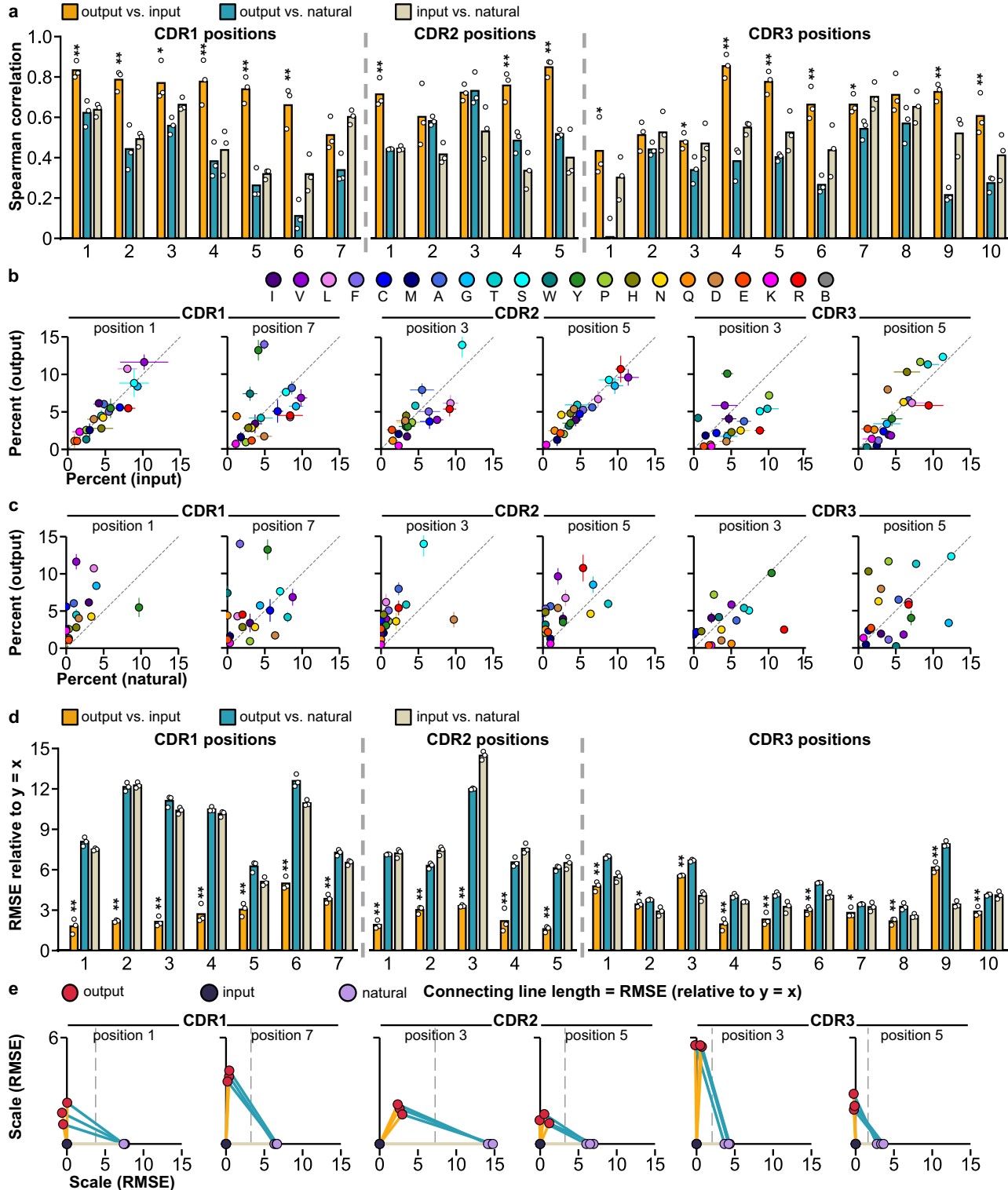

performance were poorly correlated across variants ($r^2 = 0.07$), as previously reported[20]. However, when considering each nanobody family separately, trends were stronger, and neutralization and affinity were more highly correlated for SR4 and SR6 nanobodies (Fig. 4e). This may be because variants within the same family share the same binding site and orientation. One intriguing hypothesis is that the slope of each nanobody family's linear trend reflects the sensitivity of the virus to the blocking of the family's binding site. A dose–response curve of selected nanobodies showed SR6c3 as the most potent neutralizer (Fig. 4f)

with an IC50 of 62.7 nM (Fig. 4g), comparable to potent SARS-CoV-2 neutralizing antibody Fab domains[21] and monoclonal antibodies[22] identified from human patients. Importantly, the original SR6 cluster contained only 679 sequences, representing 0.67% of the 101,674 sequences from the initial selection output, highlighting the power of CeVICA in rapidly identifying high-performance antibodies among a vast number of potential candidates.

Next, we examined the potential impact that our nanobody sequences may have on immunogenicity in humans, as a major

**Fig. 3 Unique output binders amino acid profiles are more similar to that of input library than natural nanobodies. a** Spearman correlation coefficient values for the amino acid percentages in the indicated sequence group pairs at each CDR position. 298 natural nanobodies (natural) and 298 randomly sampled sequences from input library (input) and output binders (output) were analyzed. Three random sampling trials were performed to generate three Spearman correlation coefficients for each position. Bar height: mean, circle: the value of each trial. **$p < 0.01$, *$p < 0.05$ (two-sided $t$ test between output vs. input and output vs. natural values, no multiple comparison adjustments). **b** Scatter plots of the percentage of each amino acid in the input library and the output binders and **c** that in the natural nanobodies and the output binders at representative CDR positions. Circle: mean, error bar: standard deviation of the three sampling trials described in **a**. A few data points are out of the range of the set axes due to extreme "outlier" values in the natural profile, see Supplementary Data 2 for all data point values. **d** Root means square error (RMSE, relative to $y = x$ line) values for the indicated sequence group pairs at each CDR position. Using the same randomly sampled sequences as **a**. Bar height: mean, circle: the value of each trial. **$p < 0.01$, *$p < 0.05$ (two-sided $t$ test between output vs. input and output vs. natural values, no multiple comparison adjustments). **e** Three-way distance maps of the distances between the three groups, with the length of each line connecting between two sequence groups indicating their RMSE. The input group (input) is fixed at (0,0), the natural group (natural) is fixed on the $x$-axis ($x$,0), and the position of the output group (output) is calculated based on its distance (RMSE) to the input and natural groups. Vertical dashed lines indicate the middle point of the distance between the input and natural groups. Source data for **a**, **d**, and **e** are provided in the Source Data file. Source data for **b** and **c** are provided in Supplementary Data 2.

concern related to the therapeutic use of nanobody is the possibility that, as camelid proteins, they would elicit an immune response. In particular, VHH hallmark residues in frame2 constitute a major difference between camelid VHHs and human VHs (Supplementary Fig. 2). We used our affinity maturation data to identify potential conversion options for these VHH hallmark residues. In three of the four VHH hallmark residues, we found nanobodies in which the residues were converted to the corresponding human residue as a result of affinity maturation (Supplementary Fig. 8, arrows). These data imply that at least some of the VHH hallmark residues can be converted to human residues without loss of binding fitness. Such conversions may serve as frame features of future nanobody library designs and improve tolerance of in vitro-engineered nanobodies by humans. Notably, single-domain antibody frames containing all four human hallmark residues have been successfully used for in vitro engineering of single-domain antibodies without light chain[23], demonstrating the feasibility of converting VHH hallmark residues to human residues. Overall, the extension of CeVICA for affinity maturation offers a strategy for improving antibody function.

**True binders and neutralizers identified across the CeVICA predicted list.** We next asked whether true binders and/or neutralizers can be identified from lower-ranked clusters across our full list of 862 clusters. To this end, we cloned and purified 24 additional nanobodies representing 24 clusters of different cluster size ranks (SR15-38, cluster size: 156 to 5, Supplementary Fig. 9). We assayed these nanobodies by both ELISA and pseudovirus neutralization against wild-type RBD/spike along with the recently emerged RBD/spike variants N501Y and E484K[24]. 19 nanobodies showed positive ELISA readings (background subtracted OD 450 nm >0.02, Supplementary Fig. 9a, b) and 5 nanobodies (SR15, 18, 25, 30, 38) had >20% inhibition at 1 μM for at least one RBD/spike variant (Supplementary Fig. 9c). Notably, SR38, representing a cluster with a cluster size of 5 that ranked at the bottom of the list of 862 clusters, binds N501Y RBD strongly and showed stronger inhibition of pseudoviruses carrying N501Y and E484K mutations compared to two previously identified nanobodies of animal origin, Ty1[25] and Nb21[26] (Supplementary Fig. 9c). Taken together, we identified 30 positive binders among a total of 38 tested nanobodies (78.9% positive rate), further validating the efficacy of the CDR-directed clustering approach for the selection of binders.

**Engineering potent nanobodies for virus neutralization.** To engineer a more potent virus-neutralizing agent, we performed a second affinity maturation using SR6c3 as the baseline template.

We identified mutation combinations that greatly enhanced binding affinity (SR6v1, SR6v7, SR6v9, and SR6v15) compared to SR6c3 (Fig. 5a, Supplementary Data 7 and 8). SR6v15, the variant with the highest binding by ELISA, had a $K_D$ of 2.18 nM as measured by biolayer interferometry (Fig. 5b) and inhibited pseudovirus infection more potently than SR6c3 (Fig. 5c). We further converted SR6v15 (SR6v15.m) into a tandemly linked dimer (SR6v15.d) or trimer (SR6v15.t), and compared them to Nb21[26]-based agents (monomer: Nb21.m, dimer: Nb21.d, trimer: Nb21.t) with pseudovirus neutralization assay (Fig. 5d). The most potent SR6v15 based agent, SR6v15.d had an IC50 of 0.329 nM, while the most potent Nb21 based agent, Nb21.t, had an IC50 of 0.244 nM (Fig. 5d). These results demonstrate CeVICA's capability to produce highly potent virus-neutralizing agents through iterative optimization.

**CeVICA selected nanobodies have good biophysical characteristics and are stable.** CeVICA uses NNB codons to randomize CDRs, which may cause over-representation of certain amino acids that could contribute to poor biophysical properties in the output nanobodies. We evaluated the extent of such potential undesired effects by several biophysical assays. First, we performed size exclusion chromatography analysis of three nanobodies (SR12, SR18, SR6c3) and found that for each of them >90% of the molecules exist as monomers (Supplementary Fig. 10). Second, we investigated the impact of cysteines in CDRs on nanobody biophysical properties and function because cysteine occurs at much higher frequencies in our library CDRs (5.8% in input library and 6.0% in unique output binders, Supplementary Data 2) than in natural CDRs (2.1% on average in CDR3 positions 7–12, Supplementary Data 2). Non-reducing SDS–PAGE gel analysis of nanobodies with 0–2 cysteines in their CDRs (using samples stored at 4 °C for at least 4 weeks) revealed that nanobodies with no CDR cysteine (SR12, SR18) only had one monomer band (Supplementary Fig. 11a), while nanobodies with 1 or 2 CDR cysteine either had a single monomer band (SR4, SR15, SR38) or a monomer and a dimer band (SR6c3, SR1, SR20, SR26) (Supplementary Fig. 11a). Analyzing SR6c3 samples that have been stored for varying lengths of time showed that dimers were not detected in freshly purified samples and appeared over time at a relatively low rate (Supplementary Fig. 11b). Thus, the presence of cysteines in CDRs did not always cause nanobody dimers due to disulfide bond formation. We next evaluated the functional consequences of CDR cysteine-mediated dimer formation. A 7-month-old SR6c3 sample showed increased signal in ELISA compared to a fresh sample, and the signal increase was suppressed by treating with a reducing agent that breaks up disulfide bonds (Supplementary Fig. 11c). The stored sample also inhibited pseudovirus infection more effectively than the fresh

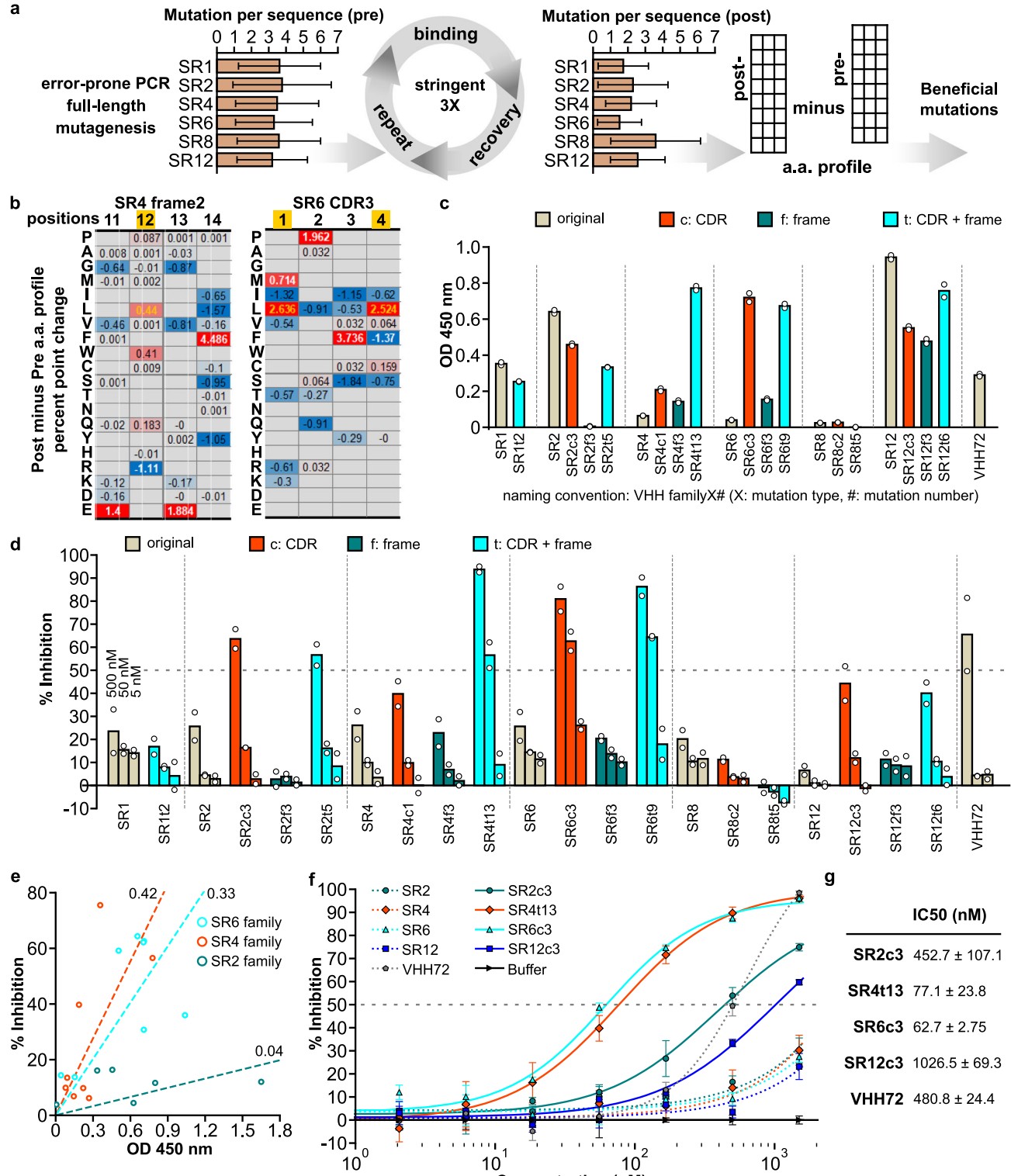

**Fig. 4 An affinity maturation strategy enhances the binding and neutralization properties of synthetic nanobodies. a** Affinity maturation workflow. **b** Two representative sections of position-wise post- minus pre-affinity maturation amino acid percent point change profile. White values indicate the original amino acid, yellow values indicate the beneficial mutation. Empty positions indicate amino acids not detected in either the pre- or post-selection libraries. **c** ELISA assay of nanobody variants. **d** SARS-CoV-2 S pseudotyped lentivirus neutralization assay of nanobodies on HEK293T expressing ACE2 and TMPRSS2. For **c** and **d**, the data shown are two technical replicates, bar height: mean, circle: the value of each replicate. **e** Scatter plot of ELISA assay absorbance versus pseudotyped lentivirus neutralization as percent infection inhibited. Nanobody concentration for both assays was 50 nM. Values are the mean of two technical replicates. Numbers on linear fitting lines were $r^2$ values for data within each family. **f** Dose–response curve for neutralization of pseudotyped lentiviral infection by nanobodies. Markers: mean of three technical replicates, error bars: standard deviation. **g** IC50 calculated from data in **f**, as mean ± standard deviation. Source data for **a**, **c**, **d**, **e**, and **f** are provided in the Source Data file.

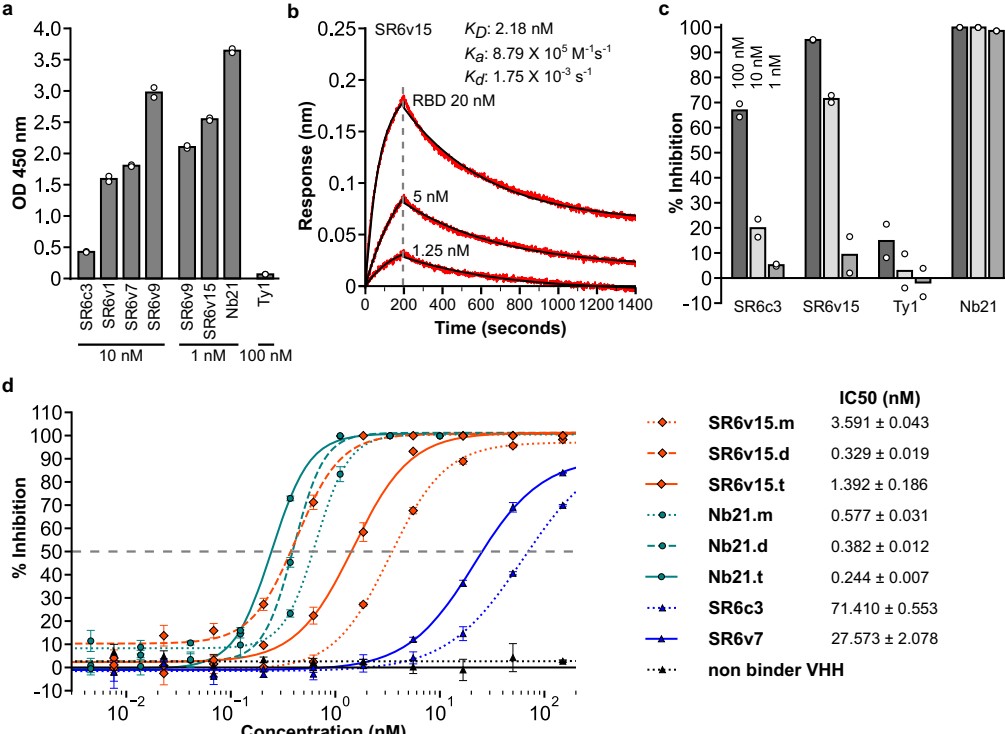

**Fig. 5 A second affinity maturation generates neutralizing agents with picomolar IC50. a** Binding (*y*-axis, ELISA assay) of SR6 variants identified by the second affinity maturation and two previously reported nanobodies, Nb21 and Ty1 (*x*-axis). Nanobody concentrations are shown at the bottom. Data shown are two technical replicates, bar height: mean, circle: the value of each replicate. **b** Biolayer interferometry assay of SR6v15. Red traces: recorded sensorgrams, black traces: fitted curves. $K_D$, $K_a$, and $K_d$ values are the mean of five measurements. **c** Pseudovirus neutralization. % inhibition (*y*-axis) of different nanobodies (*x*-axis). Data shown are two technical replicates, bar height: mean, circle: the value of each replicate. **d** Dose–response curve for neutralization of pseudotyped lentiviral infection by nanobodies and nanobody-based agents. Markers: mean of three technical replicates, error bars: standard deviation. IC50 values are shown as mean ± standard deviation. Source data for **a**, **c** and **d** are provided in the Source Data file.

sample (Supplementary Fig. 11d), consistent with ELISA data and indicating that disulfide bond formation via CDR cysteine does not adversely affect the function of SR6c3.

Finally, we assessed the thermal stability of nanobodies produced by CeVICA. Both SR6c3 and SR6v15 showed good resistance to thermal denaturation and had a melting temperature of 72 °C (Supplementary Fig. 12a), which is comparable to nanobodies generated by other methods[26]. We then tested the ability of different nanobodies to refold after complete thermal denaturation by comparing ELISA readings of nanobody samples before and after heating at 98 °C for 10 min. SR6v15 showed a higher refolding with a heated/nonheated ratio of 0.72 compared to that of VHH72[19] (0.33) and Nb21[26] (0.57) (Supplementary Fig. 12b). Surprisingly, SR6c3 had a heated/nonheated ratio >1, indicating increased binding affinity after complete thermal denaturation and refolding. We hypothesize that this increase may result from expedited disulfide bond formation that increased the percentage of dimers in SR6c3 samples subjected to heating. This hypothesis is supported by the observation that SR6c3 samples heated and refolded in the presence of reducing reagent had a heated/nonheated ratio of 1 (Supplementary Fig. 12b). Thus, nanobodies produced by CeVICA have good thermal stability and can be efficiently refolded after complete thermal denaturation.

## Discussion

CeVICA offers a generalizable solution for in vitro nanobody engineering that integrates all the components necessary to generate nanobody binder sequences in a cell-free process (Fig. 1a–f). The CeVICA nanobody library was designed to contain only the essential features for robust nanobody structure, revealed by the diversity profile across the length of natural nanobodies (Supplementary Fig. 1a, c). We validated that fully random NNB encoded codons in all CDR positions do not adversely affect binder selection (Fig. 3) nor impact biophysical stability of individual nanobodies produced by the platform (Supplementary Figs. 10–12). A linear DNA library of this design can be efficiently produced by successive PCR and ligation (Fig. 1g), yielding large libraries whose size can be directly quantified. When using oligos containing alternative base mix ratios the same process can yield different amino acid profiles for specific CDR positions, and alternative frame template sequences can be used to enrich for unique biophysical properties encoded in the frame regions of nanobodies. Moreover, these linear libraries performed well when used as input to an optimized ribosome display-based selection protocol, which suppresses sequence segment shuffling that could break up CDR pairing (see the "Methods" section), a challenging problem previously associated with cell-free systems[18].

A key feature of CeVICA is binder sequence recovery using CDR-directed clustering. This approach fully utilizes all sequences in the output library to provide a comprehensive view of all binders contained in the output (Figs. 1d–f, and 2b–e) and, in effect, reduces the nanobody characterization screen space (e.g., From 19,223 nanobody sequences in RBD output library to 862 in the list of nanobody clusters) (Fig. 2b, d, e). As a result, CeVICA is particularly suited for applications where large numbers of antibodies need to be screened to isolate ones with unique traits in specialized assays (e.g., virus neutralization, receptor activation, targeting hard-to-target epitopes) in addition to target binding.

Indeed, when we applied CeVICA to engineer SARS-CoV-2 neutralizing nanobodies, we were able to identify SR38, a nanobody with a rare ability to strongly favor binding of N501Y containing RBD and to neutralize N501Y containing pseudovirus more potently than pseudovirus not carrying N501Y (Supplementary Fig. 9). SR38 is thus a potential candidate for the development of N501Y variant-specific detection reagents or cross-variant neutralization agents. Importantly, SR38's cluster only contained 5 sequences, representing ~0.03% of the total, making it difficult to recover by random sampling without computational clustering.

Previous synthetic nanobody library designs sought to randomize CDR positions using an amino acid profile that recapitulates the profile observed in the corresponding positions in natural nanobodies; however, to the best of our knowledge, whether the natural profile represents an ideal profile for the purpose of in vitro antibody engineering has not been thoroughly investigated experimentally. The large number of nanobody clusters we generated using CDR-directed clustering offered the opportunity to test the fitness of randomized amino acid profiles in binder selection (Fig. 3). We found that in many positions, the output binder profile highly resembles the input library profile, while the similarity between the output profile and the natural profile is lower. For positions where the output profile moved significantly away from the input profile (e.g., CDR1 position 7), the distance between output and natural profiles is greater than that between output and input, and also greater than the distance between input and natural, indicating that the output profile is not moving closer to a natural profile in these positions (Fig. 3d, e). Thus, we did not find evidence indicating that the amino acid profiles observed in natural nanobodies are more fit than an NNB profile for binder selection (although they may be more fit for other features). These data also suggest a strategy to improve the fitness of an input library by incorporating amino acid profiles that match the output profile, which can be achieved by using specifically defined (non-equal) base mix ratios for the three base positions of a randomizing codon. Such a strategy could provide future improvements in synthetic nanobody library design.

The nanobodies produced by CeVICA showed good biophysical properties that are comparable to nanobodies of animal origin (Supplementary Figs. 10–12). Notably, there was robust refolding after complete thermal denaturation up to 100% (SR6c3) (Supplementary Fig. 12b). Such high refolding capability may be partly explained by the use of ribosome display for the selection of these nanobodies, during which nanobodies must fold into their functional confirmation while tethered to ribosomes in a minimally reconstituted protein synthesis environment that lacks factors, such as chaperones, normally found inside cells to aid protein folding, thus enriching for nanobodies with strong inherent folding stability. This hypothesis could be tested further in the future when CeVICA is applied to more cases. The most potent nanobody generated in this study, SR6v15, outperformed two of the previously reported nanobodies generated through animal immunization, Ty1[25] and VHH72[19], in both the binding affinity to RBD and the neutralization potency of pseudovirus infection (Fig. 5a, c). A dimeric form of SR6v15, SR6v15.d, had a more than 10-fold increase in pseudovirus neutralization potency compared to the monomeric SR6v15. SR6v15.d's IC50 is comparable to that of Nb21.t, a previously reported nanobody with high virus neutralization potency[26] (Fig. 5d). Taken together, these data demonstrate CeVICA's suitability for engineering high-affinity nanobodies with comparable biophysical properties as nanobodies produced by animals, making it a valuable addition to in vitro antibody engineering technologies. Given its seamlessly integrated procedure, CeVICA is amenable to automation and could provide an important tool for antibody generation in a rapid, reliable, and scalable manner. CeVICA further provides a technology framework for the incorporation of future refinements that could overcome limitations of in vivo fitness of in vitro generated antibodies and the overall efficiency of cell-free antibody engineering.

## Methods

**Constructs.** DNA encoding nanobodies were obtained by gene synthesis (IDT) and cloned into a pET vector in frame with a C-terminal 6xHis tag or GST tag by Gibson assembly (NEBuilder® HiFi DNA Assembly Master Mix, New England Biolabs). DNA encoding SARS-CoV-2 S RBD (S a.a. 319–541) was obtained by gene synthesis and cloned into pcDNA3 with an N-terminal SARS-CoV-2 S signal peptide (S a.a. 1–16) and a C-terminal 3xFlag tag by Gibson assembly. EGFP was cloned into pcDNA3 with a C-terminal 3xFlag tag by Gibson assembly. SARS-CoV-2 S was amplified by PCR (Q5 High-Fidelity 2x Master Mix, New England Biolabs) from pUC57-nCoV-S (a kind gift from Jonathan Abraham lab). SARS-CoV-2 S was deleted of the 27 a.a. at the C-terminal and fused to the NRVRQGYS sequence of HIV-1, a strategy previously described for retroviruses pseudotyped with SARS-CoV S[27]. Truncated SARS-CoV-2 S fused to gp41 was cloned into pCMV by Gibson assembly to obtain pCMV-SARS2ΔC-gp41. psPAX2 and pCMV-VSV-G were previously described[28]. pTRIP-SFFV-EGFP-NLS was previously described[29] (a gift from Nicolas Manel; Addgene plasmid # 86677; http://n2t.net/addgene:86677; RRID:Addgene_86677). cDNA for human TMPRSS2 and Hygromycin resistance gene was obtained by synthesis (IDT). pTRIP-SFFV-Hygro-2A-TMPRSS2 was obtained by Gibson assembly.

**Cell culture.** HEK293T cells were cultured in DMEM, 10% FBS (ThermoFisher Scientific), PenStrep (ThermoFisher Scientific). HEK293T ACE2 was a kind gift from Michael Farzan. HEK293T ACE2 cells were transduced with pTRIP-SFFV-Hygro-TMPRSS2 to obtain HEK293T ACE2/TMPRSS2 cells. The transduced cells were selected with 320 µg/ml of Hygromycin (Invivogen) and used as a target in SARS-CoV-2 S pseudotyped lentivirus neutralization assays. Transient transfection of HEK293T cells was performed using TransIT®-293 Transfection Reagent (Mirus Bio).

**Amino acid profile construction and analysis of natural nanobodies.** Nanobody protein sequences were downloaded from the Protein Data Bank (www.rcsb.org, date 2020-09-02, Supplementary Data 1) or abYsis (www.abysis.org/abysis, date 2021-05-01, Supplementary Data 1). Nanobodies were separated into CDRs and frames (segments) by finding regions of continuous sequence in each nanobody that best matched the following standard frame sequences:

 frame1 standard: EVQLVESGGGLVQAGDSLRLSCTASG,
 frame2 standard: MGWFRQAPGKEREFVAAIS,
 frame3 standard: AFYADSVRGRFSISADSAKNTVYLQMNSLKPEDTAVYYCAA,
 frame4 standard: DYWGQGTQVTVSS,

Each matched region is the corresponding frame of the nanobody, the region between frame1 and frame2 is CDR1, the region between frame2 and frame3 is CDR2, the region between frame3 and frame4 is CDR3 (Fig. 1g). Only nanobody sequences with at least one unique CDR were selected to represent natural nanobodies and used for constructing amino acid profiles (a.a. profile). 298 sequences from Protein Data Bank (PDB298) and 1030 sequences from abYsis (abYsis1030) fit this selection criterion (Supplementary Data 1). The amino acid (a.a.) profile at each position within each segment was calculated by finding the percentage of each of the 20 universal proteinogenic amino acids at that position among all selected nanobodies, all frame lengths were set to the same length as frame standards. CDR lengths were set to accommodate different CDR lengths, CDR1 and CDR2 lengths were set to 10, CDR3 length was set to 30. Nanobodies with CDR lengths shorter than the corresponding set length had their CDR filled from the C-terminal end with empty position holders up to the set length. Numbers in the amino acid profile table are the percentage of each amino acid. CDR boundaries were defined by the position where the combined frequency of the top two most abundant amino acids dropped sharply.

We compared our annotation method to Kabat and Chothia annotation (www.abysis.org/abysis/sequence_input/key_annotation/key_annotation.cgi) and found all three methods (Kabat, Chothia, and ours) showed frame regions with the same core sequence, and with 1–2 amino acid differences in the exact CDR boundaries between the three methods. The performance of our library suggests our annotation faithfully captured the domain structure of nanobodies.

We used the 1-Gini index to measure the level of diversity at each amino acid position. The Gini index measures the degree of inequality among individuals in a population, ranging from 0, when resources are uniformly (equally) distributed across individuals, and 1 when one member has all the resources. Our diversity index of 1-Gini takes 0 when there is no diversity (one amino acid has an abundance of 100%) and 1 for the highest diversity (all amino acids have the same abundance). The diversity index is calculated for 8 positions for CDR1, 6 positions for CDR2, and 18 positions for CDR3 for all sequence groups, when no sequence in the group contains a certain CDR position, the diversity index will be 0. For example, in CDR2, both the natural nanobody collection and our input library

contained a very small percentage of nanobodies having CDR2 with 6 a.a., while the output binder collection has no nanobody having CDR2 with 6 a.a., hence the diversity index has a value of 0 for the output binder plot in Supplementary Fig. 6b but a non-zero value for natural nanobodies and input library in Supplementary Fig. 1c, d.

**Nanobody library design and construction**. Nanobody library sequence is designed to recapitulate the sequence diversity of frames and CDRs observed from analyzing natural nanobodies (PDB298, abYsis1030, Supplementary Data 2). Our design differs from prior designs[6–8] in both the length of CDRs, the positions selected for randomization, and the randomization strategy. Such differences likely arise from differences in the size of natural nanobody collection retrieved from databases (93 in McMahon et al. [6] versus 298/1030 in this study) and/or in how the nanobodies are annotated and analyzed ("amino acid profile construction and analysis of natural nanobodies"). For example, our analysis showed the percentage of nanobodies containing CDR2 with lengths 4, 5, or 6 amino acids (a.a.) are 32%, 61%, and 1.7% respectively, we thus chose to use CDR2 with a length of 5 a.a. to recapitulate the most prevalent CDR2 length. In contrast, McMahon et al. [6] used an equivalent CDR2 length of 4 a.a., while Moutel et al. [7] used an equivalent length of 6 a.a. (Supplementary Fig. 4).

Nanobody libraries were constructed by ligation of PCR products in three stages, with each stage randomizing one of the three CDRs. Primers used and PCR cycling conditions for each primer pair are listed in Supplementary Data 3. Primers were synthesized by IDT (www.idtdna.com) using the standard DNA oligo synthesis and purified by desalting without PAGE purification, we find the level of synthesis errors with standard oligo synthesis and desalting purification do not have a significant impact on the functionality of the nanobody library. At each stage, PCR was performed using a high-fidelity DNA polymerase without strand displacement activity (Phusion High-Fidelity DNA Polymerase, New England Biolabs). Importantly, 65 °C was used as the elongation temperature to avoid hairpin opening during DNA elongation. PCR products with the correct size were purified by DNA agarose gel extraction using NucleoSpin Gel and PCR Clean-Up Kit (Takara, this kit was used for all DNA agarose gel extraction steps in this study). Ligation and phosphorylation of PCR products were performed simultaneously using T4 DNA ligase (New England Biolabs) and T4 Polynucleotide Kinase (New England Biolabs). Ligation products with the correct size were purified by DNA agarose gel extraction. Purified ligation products were quantified with Qubit 1× dsDNA HS Assay Kit (ThermoFisher Scientific, this kit was used for all Qubit measurements in this study) using Qubit 3 Fluorometer.

CDR2 was randomized in stage one, PCR templates at this stage were equal molar mixtures of plasmids carrying DNA encoding frames, including three frame1 versions, one frame2, three frame3 versions, and one frame4. The three versions of frame1 and frame3 were derived from consensus sequence extracted from natural nanobody a.a. profile, the A3 nanobody[10] and a GFP-binding nanobody[16]. Amino acid sequences of the frames are shown in Supplementary Fig. 2.

CDR1 was randomized in stage two, 200 ng of ligation product from the first stage were digested by Not I-HF (New England Biolabs) and heat-denatured, the entire digestion product was used as a template for PCR in stage two. The ligation product of stage two was subjected to one round of ribosome display and anti-Myc selection (described in "In vitro selection"), the entire recovered RNA was reverse transcribed and PCR amplified and purified.

270 ng of this RT-PCR product was used as a template for PCR in stage three to randomize CDR3. The ligation product of stage three was purified by DNA agarose gel extraction. The purified ligation product was then digested by DraI (New England Biolabs) and a fragment of ~680 bp in size was purified by DNA agarose gel extraction to obtain the final nanobody library, referred to as the input library.

**High throughput full-length sequencing of nanobody library**. Sequencing libraries from nanobody DNA libraries were prepared by two PCR steps using primers and PCR cycling conditions listed in Supplementary Data 3. Equal mixtures of Phusion High-Fidelity DNA polymerase (New England Biolabs) and Deep Vent DNA polymerase (New England Biolabs) were used for both PCRs to ensure efficient amplification. PCR cycle number was chosen to avoid over-amplification and typically falls between 5 and 15.

In the first PCR, Illumina universal library amplification primer binding sequence and a stretch of variable lengths of random nucleotides were introduced to the 5′ end of library DNA. And similarly, Illumina universal library amplification primer binding sequence and a stretch of variable lengths of index sequence are introduced to the 3′ end of library DNA. Eight different lengths were used for both random nucleotides and index to create staggered nanobody sequences in the sequencing library, this arrangement is required for high-quality sequencing of single amplicon libraries on an Illumina MiSeq instrument. The product of the first PCR was purified by column clean-up using NucleoSpin Gel and PCR Clean-Up Kit and the entire sample was used as the template for the second PCR.

In the second PCR, Illumina universal library amplification primers were used to generate a sequencing library. Sequencing libraries were purified by DNA agarose gel extraction, quantified using Qubit 3 Fluorometer, and sequenced on an Illumina Miseq instrument using MiSeq Reagent Nano Kit v2 (500-cycles)

(Illumina, MS-103-1003), no PhiX control library spike-in was used. The sequencing run setup was: paired-end 2 × 258 with no index read. Index in the library was designed as an inline index, thus a separate index read was not required. Raw reads were separated by index, trimmed to remove N bases and bases with a quality score of <10 prior to downstream analysis.

**Ribosome display**. Nanobody DNA library containing a specified amount of diversity was first amplified using a DNA recovery primer pair listed in Supplementary Data 3. Equal mixtures of Phusion High-Fidelity DNA polymerase (New England Biolabs) and Deep Vent DNA polymerase (New England Biolabs) were used for the PCR. PCR cycle number was chosen to avoid over-amplification and typically falls between 5 and 15. In a standard preparation, 200–500 ng of the purified PCR product was used as DNA template in 25 μl of coupled in vitro transcription and translation reaction using PURExpress In Vitro Protein Synthesis Kit (New England Biolabs). The reaction was incubated at 37 °C for 30 min, then placed on ice, and 200 μl ice-cold stop buffer (10 mM HEPES pH 7.4, 150 mM KCl, 2.5 mM MgCl₂, 0.4 μg/μl BSA (New England Biolabs), 0.4 U/μl SUPERase•In (ThermoFisher Scientific), 0.05% TritonX-100) was then added to stop the reaction. This stopped ribosome display solution was used for binding to immobilized protein targets during in vitro selection. The amount of DNA template, the volume of coupled in vitro transcription and translation reaction, and the volume of stop buffer was scaled proportionally when different volumes of stopped ribosome display solution where needed. 1–8× standard preparations were used for each selection round with the first round using 8× standard preparations, the second round using 2× standard preparations, and the third-round using 1× standard preparation.

**In vitro selection**. Target proteins were immobilized to magnetic beads by first coating protein G magnetic beads (ThermoFisher Scientific, 10004D) with anti-Flag antibody (Sigma-Aldrich, F1804, at 1:50 dilution), then incubating antibody-coated beads with cell lysate or cell media containing 3×Flag tagged target proteins at 4 °C for 2 hours. For anti-Myc selection, magnetic beads were coated by anti-Myc antibody (ThermoFisher Scientific, 13-2500, at 1:50 dilution) only. 100 μl of antibody-coated beads were used for target immobilization and pre-clearing in the first round, and 50 μl were used for subsequent rounds. The beads were washed three times with PBST (PBS, ThermoFisher Scientific, with 0.02% TritonX-100). Stopped ribosome display solutions were first incubated with antibody-coated beads (without targets) at 4 °C for 30 minutes for pre-clearing of non-specific and off-target binders, the solution was then transferred to target immobilized beads and incubated at 4 °C for 1 hour, the target immobilized beads were then washed four times with wash buffer (10 mM HEPES pH 7.4, 150 mM KCl, 5 mM MgCl₂, 0.4 μg/μl BSA (New England Biolabs), 0.1 U/μl SUPERase•In (ThermoFisher Scientific), 0.05% TritonX-100). After washing, beads were resuspended in TRIzol Reagent (ThermoFisher Scientific, 15596026), and RNA was extracted from the beads, 25 μg of linear acrylamide (ThermoFisher Scientific, AM9520) were used as co-precipitant during RNA extraction. Reverse transcription of extracted RNA was performed using Maxima H Minus Reverse Transcriptase (ThermoFisher Scientific) and primer as described in Supplementary Data 3, row 64. The reverse transcription reaction was purified using SPRIselect Reagent (Beckman Coulter) to obtain purified cDNA. Purified cDNA was amplified by PCR using equal mixtures of Phusion High-Fidelity DNA polymerase and Deep Vent DNA polymerase. PCR cycle number (Supplementary Data 3) was chosen to avoid over-amplification and typically falls between 10 and 25. This PCR condition ensures efficient full-length product synthesis at each cycle and is required to faithfully amplify nanobody genes without CDR shuffling, a phenomenon[18] that could otherwise cause selection failure. The PCR product was purified by DNA agarose gel extraction. The purified PCR product was used for library generation for high throughput full-length sequencing or as DNA input for ribosome display reaction (coupled in vitro transcription and translation) to perform additional rounds of in vitro selection.

One round of anti-Myc selection was performed on the nanobody library with CDR1 and 2 randomized to enrich for correct-frame sequences. Several factors can in principle contribute to the presence of out-of-frame sequences after anti-Myc selection: (1) non-specific binding of RNA or protein to magnetic beads; (2) translation through alternative start codons downstream of areas containing out-of-frame errors; and/or (3) inefficient binding of the anti-Myc antibody to the expressed Myc peptide that is located between the VHH protein and ribosome. We disfavor (1), because although our *input* library contained 27.5% full-length sequences, the remaining sequences that contained errors do not interfere with full-length sequences and are reduced to <10% after three rounds of RBD selection (Fig. 2c), suggesting that these erroneous sequences or their encoded peptides do not non-specifically stick to beads at significant levels to impact binder selection.

As a control experiment to demonstrate the efficiency of our ribosome display and selection protocol, SR6c3 sequence was linked with 5′ and 3′ sequence elements for ribosome display and serves as control input DNA, 100 ng of control input DNA was displayed by ribosome display in a reaction volume of 10 μl and bound to 500 μl RBD-coated beads, washed and total RNA was extracted from the beads. 7910 ng total RNA was recovered, of which 989 ng is estimated to be SR6c3 RNA (1/8 of the total, calculated by the mass ratio of nanobody RNA, 649 nt, to *E. coli* ribosomal RNAs, 4568 nt), representing a coverage rate of 19× in the output.

**CDR-directed clustering analysis**. Computational analysis for CDR-directed clustering was performed using custom python scripts. Paired-end sequences were merged into full-length nanobody sequences. Merged nanobody sequences were quality trimmed and translated into nanobody protein sequences, which were separated into CDRs and frames (segments) as described in the "Amino Acid Profile Construction and analysis of natural nanobodies" section. Two nanobodies were determined to have similar CDRs via the following steps. First, the ungapped sequence alignment score (match score) was calculated for each CDR of the two nanobodies as the sum of BLOSUM62[30] amino acid pair scores at each aligned position (if two CDRs have different lengths, their sequence alignment score was set to −5). The alignment scores of any two CDRs were summed to yield three scores, and if at least one of the three was larger than 35 (Fig. 2b), the two nanobodies were defined as having similar CDRs. Next, nanobodies with similar CDRs were grouped into a cluster by a two-step process. In the first step, we chose as nanobody cluster-forming "seeds" those nanobodies that were called similar to at least 5 other nanobodies (all remaining nanobodies were not considered for clustering). In the second step, we iteratively selected a seed nanobody with at least 5 other similar (>35 match score) seed nanobodies, and grouped all of them into one cluster, removing them from the seed nanobody pool, and iterated this procedure until no seed nanobodies remained. For RBD, there were 83,433 seeds in the first step, and 83,392 were grouped in clusters in the second step. For EGFP, 71,210 of 71,220 seeds were grouped in clusters (Supplementary Data 9). This heuristic was fast in a standard computing environment with multiprocessing capabilities.

A representative sequence to illustrate each CDR in each cluster was chosen as the most frequent CDR sequence in the cluster (the chosen representatives for CDR1, 2, and 3 may not necessarily be from the same sequence, and are used only for illustrative purposes for each cluster as in Supplementary Data 4 and 5; whole nanobody sequences were used for gene synthesis and all downstream experiments). A consensus sequence was generated for each CDR, where each position in the CDR was represented by a six-character string, such that the first and fourth characters were the single letter code for the top and the second most abundant amino acid at the position, respectively, and the following two characters (second and third for the most abundant; fifth and sixth for the second most abundant), were their frequency, respectively (ranging from 00 for <34% to 99 for 100%). The consensus sequence for a CDR was recorded as a single "B00" when the standard deviation of the lengths of all CDRs was >1. CDR scores were calculated by summing a score for each position in the CDR consensus sequence, with scores of 3, 2, 1 for positions where the most abundant amino acid had frequencies >80%, 50%, or less, respectively, and a score of 0 for CDRs with a consensus sequence of a single "B00" (Supplementary Data 4 and 5). Representative whole nanobody sequence for each cluster was selected as the one with the maximal sum (max-sum) of all CDR similarity scores between the nanobody and all other nanobodies in the cluster. This max-sum representative nanobody sequence selection process minimizes the impact of random errors introduced during NGS library preparation and sequencing by imposing a scoring penalty on sequences containing random errors.

**Protein expression and purification**. Target proteins used for in vitro selection and ELISA were prepared by transiently transfecting HEK293T cells with plasmids carrying either spike RBD with C-terminal 3×Flag tag and N-terminal signal peptide of the spike (RBD-3×Flag), or EGFP with C-terminal 3×Flag tag (EGFP-3×Flag). Cell culture media (for RBD-3×Flag) or lysate of cell pellet (for EGFP-3×Flag) was used for coating magnetic beads (for CeVICA) or plates (for ELISA). Nanobodies with C-terminal 6xHis tag (Nanobody-6xHis) were purified by expressing in *E. coli*, followed by purification using HisPur Cobalt Resin (ThermoFisher Scientific, 89964). Briefly, Nanobody-6xHis plasmids were transformed into T7 Express *E. Coli.* (New England Biolabs), single colonies were transferred into 10 ml LB media and grown at 37 °C for 2–4 h (until OD reached 0.5–1), the culture was chilled on ice, then IPTG was added to a final concentration of 10 μM. The culture was then incubated on an orbital shaker at room temperature (RT) for 16 hours. Bacterial cells were pelleted by centrifugation and lysed in B-PER Bacterial Protein Extraction Reagent (ThermoFisher Scientific) supplemented with rLysozyme (Sigma-Aldrich), DNase I (New England Biolabs), 2.5 mM MgCl₂, and 0.5 mM CaCl₂. Bacterial lysates were cleared by centrifugation and mixed with wash buffer (50 mM sodium phosphate pH 7.4, 300 mM sodium chloride, 10 mM imidazole) at a 1:1 ratio, and then incubated with 40 μl HisPur cobalt resin for 2 hours at 4 °C. The resins were then washed four times with wash buffer. Proteins were eluted by incubating resin in elution buffer (50 mM sodium phosphate pH 7.4, 300 mM sodium chloride, 150 mM imidazole) at RT for 5 minutes. Purified protein samples were quantified by measuring absorbance at 280 nm on a Nano-Drop spectrophotometer.

**ELISA assay for nanobody binding to RBD**. Maxisorp plates (BioLegend, 423501) were coated with 1 μg/ml anti-Flag antibody (Sigma Aldrich, F1804) in coating buffer (BioLegend, 421701) at 4 °C overnight. Plates were washed once with PBST (PBS, ThermoFisher Scientific, with 0.02% TritonX-100), a 1:1 mixture of HEK293T cell culture media containing secreted RBD-3xFlag and blocking buffer (PBST with 1% nonfat dry milk) was added to the plates and incubated at room temperature (RT) for 1 hour. RBD coated plates were then blocked with blocking

buffer at RT for 1 hour. Plates were washed twice with wash buffer and purified Nanobody-6xHis diluted in blocking buffer were added to the plates and incubated at RT for 1 hour. Plates were washed three times with wash buffer, HRP conjugated anti-His tag secondary antibody (BioLegend, 652503) diluted 1:2000 in blocking buffer was then added to the plates and incubated at RT for 1 hour. Plates were washed three times with wash buffer and TMB substrate (BD, 555214) was added to the plate and incubated at RT for 10–20 minutes. Stop buffer (1 N sulfuric acid) was added to the plates once enough color developed. Quantification of plates was performed by measuring absorbance at 450 nm on a BioTek synergy H1 microplate reader using Gen5 software 1.11.5. Data reported were background subtracted. Two levels of background subtraction were performed: (1) subtracting absorbance measured from wells incubated with blocking buffer only (without purified Nanobody-6xHis) from sample measurements (reflecting background absorbance by plates); and (2) subtracting absorbance from each nanobody incubated wells coated only with anti-Flag antibody and without RBD (reflecting non-specific binding of each nanobody).

**Pseudotyped SARS-CoV-2 lentivirus production and lentivirus production for transductions**. Lentivirus production was performed as previously described[28]. Briefly, HEK293T cells were seeded at $0.8 \times 10^6$ cells per well in a six-well plate and were transfected the same day with TransIT-293 Transfection Reagent and a mix of DNA containing 1 μg psPAX, 1.6 μg pTRIP-SFFV-EGFP-NLS, and 0.4 μg pCMV-SARS2ΔC-gp41. The medium was changed after overnight transfection. SARS-CoV-2 S pseudotyped lentiviral particles were collected 30–34 hours post-medium change and filtered on a 0.45 μm syringe filter. To transduce HEK293T ACE2 the same protocol was followed, with a mix containing 1 μg psPAX, 1.6 μg pTRIP-SFFV-Hygro-2A-TMPRSS2, and 0.4 μg pCMV-VSV-G.

**SARS-CoV-2 S pseudotyped lentivirus neutralization assay**. The day before the experiment, $5 \times 10^3$ HEK293T ACE2/TMPRSS2 cells per well were seeded in 96-well plates in 100 μl. On the day of lentivirus harvest, SARS-CoV-2 S pseudotyped lentivirus was incubated with nanobodies or nanobody elution buffer in 96-well plates for 1 hour at RT (100 μl virus + 50 μl of nanobody at appropriate dilutions). The medium was then removed from HEK293T ACE2/TMPRSS2 cells and replaced with 150 μl of the nanobody plus pseudotyped lentivirus solution. Wells in the outermost rows of the 96-well plate were excluded from the assay. After overnight incubation, the medium was changed to 100 μl of fresh medium. Cells were harvested 40–44 hours post-infection with TrypLE (Thermo Fisher), washed in medium, and fixed in FACS buffer containing 1% PFA (Electron Microscopy Sciences). Percentages of GFP positive cells were quantified on a Cytoflex LX (Beckman Coulter) and data were analyzed with FlowJo. During the development of the pseudotyped lentivirus neutralization assay, we found HEK293T ACE2/TMPRSS2 cells were highly susceptible to pseudovirus infection and produced consistent inhibition measurements, while Vero E6 and Caco-2 cells showed lower susceptibility in our GFP detection-based assays.

**Affinity maturation**. Error-prone PCR was used to introduce random mutations across the full length of selected nanobody DNA sequences. 0.1 ng of plasmid carrying each selected nanobody DNA sequence were used as template in PCR reactions using Taq DNA polymerase with reaction buffer (10 mM Tris–HCl pH 8.3, 50 mM KCl, 7 mM MgCl₂, 0.5 mM MnCl₂, 1 mM dCTP, 1 mM dTTP, 0.2 mM dATP, 0.2 mM dGTP) suitable for causing mutations in PCR products. Mutagenized library (pre-affinity maturation) for input to CeVICA was made by ligating PCR products of error-prone PCR that carries nanobodies to DNA fragments containing the remaining elements required for ribosome display. Three rounds of ribosome display and in vitro selection were performed on the mutagenized library as described in the "In vitro selection" section, during which the incubation time of the binding step was kept between 5 seconds and 1 minute to impose a stringent selection condition, additional error-prone PCR was not performed between selection rounds. The output library (post-affinity maturation) was sequenced along with the pre-affinity maturation library as described in the "high throughput full-length sequencing of the nanobody library" section.

**Identification and ranking of beneficial mutations**. To identify potential beneficial mutations for each selected nanobody, we built an amino acid profile (a.a. profile) table for each nanobody family in the pre- and post-affinity maturation library and identified amino acids with increased frequency in the post-affinity maturation population compared to their pre-maturation frequency. For each nanobody parental sequence, an a.a. profile was built of the percent of each a.a. across all nanobody sequences originated from one parental nanobody in the pre-affinity maturation library ("pre-a.a. profile") and in the post-affinity maturation library ("post-a.a. profile"). A percent point change table was generated by subtracting the pre-a.a. profile from the post-a.a. profile, describing the change of frequency of each observed amino acid at each position of the nanobody protein following affinity maturation.

We defined a putative beneficial mutation and assigned beneficial mutation score as either (1) the non-parental amino acid with the biggest increase in

frequency if its increase is at least 0.5 percentage points; the score is the difference from the parental amino acid frequency; or (2) the non-parental amino acid with the biggest increase after the parental amino acid if the increase is at least 1.5 percentage points; the score is the percentage point change of the beneficial mutation. To avoid too many proximal putative beneficial mutations (which may cause structural incompatibility), a putative beneficial mutation was discarded if it (1) is outside the CDRs; (2) is <3 positions away from another beneficial mutation ("nearby mutation) and has a smaller beneficial mutation score than the nearby mutation; and (3) co-occurs less than twice with the nearby mutation. From this final list of putative beneficial mutations, different combinations were chosen and incorporated into each nanobody parental sequence that includes one combination of all beneficial mutations in CDRs, one combination of the top-3 ranked (by beneficial mutation score) mutations in frames, and at least one combination of both CDR mutations and frame mutations (Supplementary Data 7).

**Biolayer interferometry**. Biolayer interferometry assays were performed on the Octet RED384 instrument (Sartorius) using anti-GST biosensors (Sartorius, 18-5096). Assays were performed in sample buffer (PBS with 0.05% Tween-20, 0.5 mg/ml BSA). Nanobodies were loaded on anti-GST biosensors in a sample buffer containing bacteria lysates of *E. coli*. expressing GST-tagged nanobodies (100-fold dilution), achieving loading levels of 1–1.2 nm. Nanobodies-loaded sensors were dipped in sample buffer containing recombinant RBD (ThermoFisher, RP-87678) for 200 seconds to record association, then dipped in sample buffer for 1200 seconds to record dissociation. Nanobody-loaded sensors dipped in sample buffer containing no RBD were used as reference sample sensors for background subtraction. No signal increase was observed for reference sample sensors which indicate no non-specific binding to loaded nanobodies. Non-specific binding of RBD to anti-GST biosensors was tested by dipping anti-GST biosensors not loaded with nanobodies in 20 nM RBD. No signal increase was observed during the incubation indicating that RBD does not bind non-specifically to anti-GST biosensors. Data analysis was performed using the Octet Data analysis software 10.0, Savitzky–Golay filtering was used to remove noise, and curves were fitted using a 1:1 binding model.

**Size-exclusion chromatography**. Size-exclusion chromatography was performed on an AKTA Pure 25M system with a Superdex increase 75 10/300 GL column (Cytiva). 50–100 μg nanobodies were loaded onto the column in running buffer (20 mM HEPES, 150 mM NaCl, PH 7.5), a flow rate of 0.5 ml/min was used and UV280 readings were recorded for 1.25 column volumes. Peak analysis was performed using the UNICORN 7 software (Cytiva).

**Thermal stability assays**. Protein thermal shift assays were performed using the Protein Thermal Shift Dye Kit (ThermoFisher, 4461146) according to the manufacturer's instructions. 4 μg of nanobodies were diluted in 1× reaction buffer and measurements were performed on a Bio-Rad CFX384 real-time PCR system using a melt curve protocol (30–98 °C, 1 °C increment, hold for 20 s then read plates using FRET channel). 98 °C heat denaturation was performed by diluting nanobody sample to 1 μM in PBS containing 100 ng/μl BSA, then heating at 98 °C for 10 min then holding at 4 °C using a PCR machine. ELISA assay of nanobody samples prior to and after complete thermal denaturation was performed as described above ("ELISA assay for nanobody binding to RBD").

**Figure plots generation**. Plots in figures were generated using python package Matplotlib 3.3.0 (https://matplotlib.org/)

**Reporting summary**. Further information on research design is available in the Nature Research Reporting Summary linked to this article.

## Data availability
Antibody sequences generated in this study are provided in Supplementary Data 7. Raw Illumina sequencing data generated in this study have been deposited in the NCBI Sequence Read Archive as a BioProject with Accession #: PRJNA756264. Natural VHH sequences used in this study are retrieved from Protein Data Bank (www.rcsb.org, date 2020-09-02, Supplementary Data 1) and abYsis (www.abysis.org/abysis, date 2021-05-01, Supplementary Data 1). Key plasmids generated in this study are deposited in Addgene. A step-by-step protocol for CeVICA is deposited in protocols.io (https://doi.org/10.17504/protocols.io.bxn9pmh6). Source data are provided with this paper in the Source Data file. Source data are provided with this paper.

## Code availability
Code for computational analysis can be accessed through Github (https://doi.org/10.5281/zenodo.5257689).

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

## Acknowledgements
We thank Christopher M. Vockley for critical reading and editing of the manuscript, Matthew H. Bakalar for helping with cloning VHH72, Leslie Gaffney and Anna Hupalowska for assistance in figure making, Michael Farzan for providing HEK293T expressing ACE2 and for discussing the SARS-CoV-2 S pseudotyped lentivirus neutralization approach, Jonathan Abraham for providing the pUC57-nCov19-S plasmid.

Work was supported by the Klarman Cell Observatory and Klarman Incubator at the Broad Institute, NHGRI 5RM1HG006193 (A.R.), HHMI (A.R.) and NIH/NIAID U19 AI082630 (N.H.). M.G. is the recipient of an EMBO Long-Term Fellowship (ALTF 486-2018) and a Cancer Research Institute/Bristol-Myers Squibb Fellow (CRI2993). Until July 31, 2020, A.R. was an Investigator of the Howard Hughes Medical Institute.

## Author contributions

X.C. and A.R. conceived the study. X.C. designed and developed the CeVICA platform, performed selection and identification of EGFP and RBD binders, performed affinity maturation of RBD binders. M.G. developed and performed SARS-CoV-2 S pseudotyped lentiviruses neutralization assay. N.H. provided support for pseudotyped lentiviruses neutralization assay. X.C. designed and performed all other experiments and analyses. X.C. and A.R. wrote the manuscript, with contributions from all co-authors.

## Competing interests

A.R. is a founder and equity holder of Celsius Therapeutics, an equity holder in Immunitas Therapeutics and until August 31, 2020 was an SAB member of Syros Pharmaceuticals, Neogene Therapeutics, Asimov and ThermoFisher Scientific. From August 1, 2020, A.R. is an employee of Genentech. N.H is an equity holder of BioNtech and is an advisor for Related Sciences. X.C. and A.R. are named co-inventors on a patent application (U.S. 63/221,663) related to CeVICA filed by the Broad Institute that is being made available in accordance with COVID-19 technology licensing framework to maximize access to university innovations. M.G. declares no competing interests.
