## [Peer Review File · Nature Communications]

Reviewers' Comments:

Reviewer #1:

Remarks to the Author:

The manuscript presented by Xun Chen, Aviv Regev and colleagues describes the generation of a synthetic VHH library, binder selections using ribosome display, NGS of input versus output libraries obtained during binder selection and a computational pipeline for CDR-directed clustering and binder identification.

The entire pipeline (called CeVICA) was applied to select synthetic nanobodies against two targets, namely EGFP and the receptor binding domain (RBD) of the spike protein of SARS-CoV-2. Some RBD nanobodies were then further characterized in terms of neutralization of pseudotyped virus entry and ELISA. Affinity maturation with error-prone PCR combined with ribosome display was performed to improve the identified nanobodies.

Input and output libraries were comprehensively compared with each other as well as with natural nanobodies, in order to determine sequence hallmarks of what the authors call "binder fitness". The paper has some clear strengths, which pertain to the bioinformatics analysis of the selection procedure, in particular the CDR-directed clustering. On the other hand, there are some major problems in the generation of the library and the manuscript falls short in convincingly showing novelty in the context of synthetic VHH library design and the use of ribosome display to enrich VHH from synthetic libraries. Another weakness is the lack of biophysical analysis of the identified RBD nanobodies.

Last but not least, the manuscript does not appropriately acknowledge recent progress in the context of synthetic VHH libraries and tries to convey the message that with this work major progress had been made to overcome seeming weaknesses of existing synthetic VHH approaches. The manuscript needs to be re-worked to highlight the major (and undoubted) advances being made at the "computational" front, which are in fact very interesting.

The manuscript text itself is well written and structured. The figures are clear and mostly self-explanatory. The excel tables provided as supplement are of high quality and are easy as well as interesting to read.

Major points

1) The authors claim that their approach to generate the synthesis VHH libraries is novel. This might be true at the technical level (namely the orientation-controlled ligation by end blocking using hairpin oligos). However, it is certainly not true for the library design principle, which in fact is based on a consensus design approach taking into account protein sequences of PDB entries of nanobodies. This was exactly the approach used by McMahon et al (Ref 8 in this paper). The only "novelty" was that the authors did randomize the CDRs with NNB codons and thus did not "bias" the amino acid composition of the CDRs to what is found in natural nanobodies (as had been done for example in Refs. 8, 11 and 13). The use of the cheaper/less sophisticated NNB primers was unlikely a very good idea (see point 2).

2) As a consequence of the NNB oligos, also undesired amino acids are introduced into the CDRs (in particular Prolines and Cysteines). As can be seen in Table S2, in particular cysteines are rarely found in natural VHHs (and if so, then in specific positions and they are then involved in a second SS-bond next to the conserved SS-bond at the centre of the VHH). In the synthetic library presented here, the frequency of cysteine is around 5 % in every randomized positions. Hence, the probability to have a synthetic VHH with at least one extra cysteine (next to the conserved pair) is around 60 % in case of the shortest CDR3 (in total 18 randomized positions for CDRs 1-3; $p = 1 - 0.95^{18}$) and around 72 % for the longest CDR3 (in total 25 randomized positions for CDRs 1-3; $p = 1 - 0.95^{25}$).

3) Of course one might argue that one does not have to care so much about these cysteines (a quick look at the RBD nanobodies presented in the paper showed that extra cysteines are found in SR1, SR4, SR6 and SR8) and also not about prolines in the CDRs, because i) they were not found to be depleted as part of the selection (see Figure S6) and the respective VHHs containing the extra-cysteines did exhibit binding and viral neutralization. However, the authors have to show non-reduced SDS-PAGE gels of the purified VHHs with binding activity against the RBD, size exclusion chromatography profiles of these VHHs, quantitative affinity measurements (BLI, SPR or similar) and finally some thermal stability data (e.g. via thermofluor using Sypro orange). This reviewer has major doubts whether the identified VHHs are monomers (because of the extra cysteines), are stable and well-behaved on size exclusion (due to "over-randomization"). Further,

the reviewer doubts whether meaningful SPR/BLI data may be recorded (again due to stickiness issues). Synthetic binders have a bad reputation because of poor biophysical behaviour. Hence, a proper biophysical analysis of synthetic binders is really very important, but is completely lacking here.

4) An important aspect of the pipeline is the NGS analysis of input and output VHH libraries. In this context, there are two technical aspects which are not touched on in the manuscript. The first one is sequencing errors. Although Illumina is pretty good when it comes to errors, these nevertheless exist. In how far did sequencing errors influence the outcome of the computational pipeline? The second problem is probably more severe. If one uses highly homologous sequences (as is the case here), PCR amplification leads to loop-shuffling (see also Fig. S3 in this paper: <https://www.nature.com/articles/s41592-019-0389-8>). The authors amplified the output/input libraries twice with PCR to introduce the Illumina sequencing adaptors. Consequently, the composition of the CDRs 1-3 of the sequences obtained are only partially the compositions that were really present in the output/input library. The authors should comment on this problem and may even tweak their computational pipeline to account for PCR-induced shuffling.

Minor points

1) Line 55: ref 11 did not produce inconsistent results, as the paper shows that for soluble proteins such as maltose binding protein, one can get really good and biophysically well-behaved binders with ribosome display alone. That paper makes the point that for challenging membrane proteins, change of display systems (ribosome display and phage display) are needed to enrich for specific VHs. The target protein used in this study (EGFP and RBD) are in fact highly stable soluble proteins, which can be considered as easy targets.

2) Line 76: Author statement: "Clustering following high throughput sequencing identifies them more efficiently than methods that rely on the analysis of individual colonies or sequences". The authors do not provide experimental evidence for this statement.

3) Line 77: Author statement: "promising a more comprehensive view of the landscape of binder potential, with minimal time and resources."
I agree with the comprehensive view, but I doubt with the time and resources statement. NGS takes time and is costly. Gene synthesis of VHs gets cheaper, but is still more costly than getting the clones from minipreps after ELISA analysis. And certainly it takes more time.

4) Line 104: the authors obviously had problems with the oligo qualities, as there seemed to be many indels leading to frameshifts/early stop codons. This is why they used this trick of displaying the VHs after randomizing CDR1 and CDR2 on the ribosome and capture via a C-terminal Myc-tag. Did the authors use PAGE-purified oligos to assemble the library? If not, this might explain why the problem was so severe. And can the authors give an explanation why the percentage of in-frame sequences did not increase more (only from 25 % to 52 %). Why were the 48 % out-of-frame VHs (which thus lack the myc-tag) captured/co-enriched as well? My suspicion: they were simply sticking to the anti-Myc antibody or the beads due to poor biophysical properties.

5) Line 136: Author statement: "The shared clusters likely target the shared components (protein G, anti-Flag antibody) present on the solid support surfaces, and thus represent background binders."
The authors do not provide experimental evidence supporting this statement.

6) Line 140: top-ranking in terms of what? Match scores of CDR1-3 summed up?

7) Line 159 and 165: "fit profile" and "fitness cost" not influenced by complete randomization: This seems to be indeed the case in terms of enrichment against the target proteins during the selection. But as outlined in the major points above, such a "reckless randomization" approach likely compromises the biophysical properties of the VHH proteins.

8) Line 177: "fitness drawbacks in vivo": this reviewer thinks that there are also major "fitness drawbacks" in vitro, namely in terms of biophysical properties etc. (see comments above).

9) Line 209: sequences instead of sequenced.

10) Line 226: This paper did not optimize the in vitro selection itself.

11) Line 339: the materials and methods lacks any downstream processing of the Illumina sequencing data. How did the authors account for sequencing errors? On what basis were sequences discarded (e.g. frameshifts, N reads, etc)?

12) Line 341 ff: the authors did not make any control experiment to show how efficient their ribosome display setup is. A good option is to display a singly high affinity nanobody on ribosomes, followed by a pull-down and quantification by qPCR.

13) Line 358 ff: the author did not state volumes and amount of beads used.

14) Line 370: what primer was used to perform reverse transcription?

15) Figure S6c, CDR1 position 1: there must be an error. The sum of the % of the 20 amino acids needs to be 100 % for both axes. But the points are too much shifted to the top/left. Hence the % sum for both axes cannot be the same, although they should.

Reviewer #2:

Remarks to the Author:

The authors describe a ribosomal display approach to select VHHs from a semi-synthetic library combined with a bioinformatic approach to identify binders. The overall strategy is very good also the preselection step on myc to increase the percentage of full length VHH binders during library construction is convincing. The affinity matured antibody SR6_c3 shows an inhibition (IC₅₀) of SARS-CoV-2 in the same range as the VH domain antibody ab8 from a phage display library (Li et al 2020). The paper also shows the need for affinity maturation of the VHH binders against RBD using this approach.

Major Revisions:

- line 35: "However, broad application of such in vitro methods remains a challenge...". I can not agree with this statement, because currently 13 FDA/EMA approved antibodies were generated by the in vitro technology antibody phage display (Alfelah et al 2020). These in vitro approaches were also used to generate neutralizing SARS-CoV-2 antibodies (Bertoglio et al 2020, Li et al 2020).
- the authors should add a negative control antibody in the neutralization assays.
- I suggest to measure the affinities which would show the effect of affinity maturation in addition to the given inhibition assays.
- I'm missing a discussion in this article on the advantage/disadvantages of other ribosomal display or in vitro selection procedures in comparison to CeVICA. I'm also missing a discussion on the generated anti-RBD VHHs.

Minor Revisions:

- 45/46: "screen diversity" > "screening diversity", "library diversity"?
- 46/47: "...DNA library delivery into cells (typically <10¹⁰)". This is not correct. The "big" naive antibody gene libraries have a library of size >10¹⁰ (e.g. Hoet et al 2005, Schofield et al 2007, Glanville et al 2009, Lloyd et al 2009, Kügler et al 2015). Or do the authors mean the transformations efficiency of E.coli which is normally in the range of 1E8
- Why did the authors use HEK293T ACE2/TMPRSS2 instead of Vero or Caco cells which are the established cell lines in SARS-CoV-2 neutralization assays?
- Please give an explanation for the the "Gini-Index"
- Please give an explanation for the "Match Score"
- why did the authors isolate 52 identical antibodies from the panning against EGF and RBD? Are these antibodies against the Flag-Tag?

Reviewer #3:

Remarks to the Author:

The authors develop a method termed CeVICA for the isolation of diverse nanobodies from a synthetic library. This method involves three parts. First, a synthetic library was prepared. This library is based on nanobodies deposited in the PDB, but incorporates large diversity through the randomization of all three CDRs with NNB codons. Second, CeVICA incorporates ribosome display and MACS for the in vitro selection of nanobodies with affinity toward a chosen antigen. Finally, sequences were clustered into families by examining sets of two sequences for pairs of identical residues in their CDRs. CeVICA then allows for analysis of nanobodies from diverse families.

This method is applied to the generation of nanobodies which bind to and neutralize SARS-CoV-2. Three rounds of selection were performed using the receptor binding domain (RBD) as the antigen. This selection strategy led to the identification of several clusters within the output VHH DNA. Fourteen VHs representing each of the top clusters were analyzed for binding affinity and neutralization activity. Of the examined nanobodies ~71% (10/14) showed some level of affinity toward the RBD in ELISA, and ~43% (6/14) showed some level of neutralization activity. The CDRs of VHs in the sorting output was also examined, and the composition of these CDRs was found to be more similar to the input library than the natural VHs contained within the PDB.

The authors then performed affinity maturation on VHs isolated from this sorting. An error prone library was generated, sorted for three rounds, and common mutations were identified. Individual clones containing selected mutations and combinations of these mutations were analyzed for binding and neutralization activity. Some nanobodies showed both increased affinity and neutralization activity while others showed only an increase in neutralization activity. Finally, the authors also observed that the error prone sorting introduced some residues consistent with humanization of camelid antibodies.

Overall issues:

1. It does not appear that this method produces nanobodies with properties that rival the best neutralizing nanobodies from other methods. The best inhibition potency is only ~60 nM. No affinity measurements (KD values are reported). The authors compare their neutralization activity to a previously reported nanobody (VHH-72), but this is a nanobody that was discovered for neutralizing SARS-CoV-1 and also neutralizes SARS-CoV-2. A more fair comparison would be to compare relative to nanobodies such as Ty1 that were developed specifically for SARS-CoV-2 (Hanke, L., Vidakovics Perez, L., Sheward, D.J. et al. An alpaca nanobody neutralizes SARS-CoV-2 by blocking receptor interaction. *Nat Commun* 11, 4420 (2020). <https://doi.org/10.1038/s41467-020-18174-5>).

2. It is unclear if the nanobodies produced by this cell free method have the stabilities and solubilities that would make them useful as reagents or therapeutics. No biophysical data is reported for melting temperatures and %monomer by size-exclusion chromatography, which prevents evaluation of how good these nanobodies are relative to previously generated nanobodies, including those generated via immunization.

Specific issues/questions:

1. The observation that, at least at several positions, the residues in the output of the sorting more closely correlates with the representation of residues in the randomly diversified input library than the natural representation of residues in nanobodies from the PDB has the potential to expand the designs of synthetic nanobody libraries available for in vitro selection. However, this observation is based on a relatively small number of sequences from the PDB (298). Would these conclusions regarding the shift away from the natural profile still hold true if a larger database, such as AbYsis, was used to look at the distribution of residues in natural VHs?

2. This paper provides a method for identifying multiple nanobodies directed toward a single antigen with diverse CDRs. It would be helpful to expand upon the advantages of enabling the selection of diverse sequences. For instance, the authors could examine whether this method has the potential to direct antibody binding toward different epitopes within the same antigen; this ability would be particularly useful for antigens in which antibodies are commonly directed toward

immunodominant epitopes. Could the authors clarify the need that they aim to address through the selection of diverse nanobody sequences?

3. "we chose one representative VHH gene from each of the 14 top-ranking RBD unique clusters and validated it for spike RBD binding and SARS-CoV-2 pseudovirus neutralization"

a. This is an interesting approach for isolating nanobodies with diverse sequences and binding properties. Would other nanobodies contained in the same cluster have similar binding and neutralization characteristics to one another? Specifically, if a nanobody is identified as having good affinity or neutralization, could other nanobodies in this cluster be examined for comparable or improved characteristics?

4. "In three of the four VHH hallmark residues there were VHHs where the residues were converted to the corresponding human residue as a result of affinity maturation (Fig. S7, arrows). These data imply that at least some of the VHH hallmark residues can be converted to human residues without loss of binding fitness."

a. Based on Fig. S7, it seems that, at most, two humanizing mutations occurred in the same sequence and there was a greater increase in VHH hallmark residues at more of the examined positions. Were individual clones with two or more such mutations (camelid to human) examined for affinity and stability in the absence of a light chain?

5. "VHHs were separated into CDRs and frames (segments) by finding regions of continuous sequence in each VHH that best matched to the following standard frame sequences:"

a. How does this methodology align with the CDR and framework definition of more standard numbering systems like Kabat or Chothia? In Table S1, some sequences have only one or two residues in CDR3 (e.g. 6SSI, 58HD, 5L21), which seems to suggest part of the CDR may be included in the framework regions using this method.

b. Additionally, could the authors clarify what is meant by "bad sequence" for several of the PDB entries in the sheet labeled "all_VHH_RCSB"?

6. "We introduced 7 random amino acids for CDR1, 5 for CDR2, and 6, 9, 10 or 13 for CDR3 to match the most commonly observed CDR lengths in natural VHHs."

a. In Figure S1, the diversity index in plots C and D show the diversity for 6 positions in CDR2. This seems to disagree with the design of 5 residues in CDR2 for the synthetic VHHs as well plot B and Table S2 which show this position as entirely blank. In Figure S5, the sixth position is again shown as entirely blank in plot A, but no diversity is listed in the diversity index in plot B. Could the authors comment on the change in number and represented diversity for CDR2?

7. "Moreover, correlation of amino acid profiles between output binders and natural VHHs are significantly less than between output binders and input library at most CDR positions (Fig. S6)."

a. In Fig. S6, are the differences in r^2 values statistically significant at any position? It seems that in CDR2, the output vs natural r^2 values are higher than the input vs output values for at least two to three positions, and in CDR3, the input vs natural values are higher than the input vs output values for at least two positions.

Response to Reviewers

Overview

We thank the Reviewers for their appreciation of our work and their thoughtful suggestions and comments. We have addressed all the points in the revised manuscript with new experiments, analyses, and substantial changes and additions to the manuscript text. We summarize the key revisions below and then address each of the Reviewers' comments in the following point by point response.

The key highlights of our revision are:

- A second affinity maturation that further increased binding affinity and virus inhibition potency of VHH (**new Fig. 5a-c**).
- Multivalent engineering that produced a neutralizing molecule with a picomolar IC50 (**new Fig. 5d**).
- Expanded testing of VHH clusters identified additional binders, with a total positive rate of 78.9%, and identified virus neutralizers that cross-neutralize different spike variants (**new Supplementary Fig. 9**).
- Consolidated our random codon fitness analysis with a larger natural VHH data set from abYsis database (**new Supplementary Fig. 7**).
- Demonstrated good biophysical properties of VHHs generated by our method showing no adverse effects from CDR cysteines (**new Supplementary Fig. 11**), good separation behavior on size-exclusion chromatography (**new Supplementary Fig. 10**) and high thermal stability (**new Supplementary Fig. 12**).
- Text revisions to provide better context of our approach in light of prior studies, such as better acknowledging progress on synthetic VHH libraries, and be clearer on the specific contributions of this study, including a focus on the computational approach.

Response to Reviewer 1

The manuscript presented by Xun Chen, Aviv Regev and colleagues describes the generation of a synthetic VHH library, binder selections using ribosome display, NGS of input versus output libraries obtained during binder selection and a computational pipeline for CDR-directed clustering and binder identification.

The entire pipeline (called CeVICA) was applied to select synthetic nanobodies against two targets, namely EGFP and the receptor binding domain (RBD) of the spike protein of SARS-CoV-2. Some RBD nanobodies were then further characterized in terms of neutralization of pseudotyped virus entry and ELISA. Affinity maturation with error-prone PCR combined with ribosome display was performed to improve the identified nanobodies. Input and output libraries were comprehensively compared with each other as well as with natural nanobodies, in order to determine sequence hallmarks of what the authors call “binder fitness”.

The paper has some clear strengths, which pertain to the bioinformatics analysis of the selection procedure, in particular the CDR-directed clustering. On the other hand, there are some major problems in the generation of the library and the manuscript falls short in convincingly showing novelty in the context of synthetic VHH library design and the use of ribosome display to enrich VHH from synthetic libraries. Another weakness is the lack of biophysical analysis of the identified RBD nanobodies.

Last but not least, the manuscript does not appropriately acknowledge recent progress in the context of synthetic VHH libraries and tries to convey the message that with this work major progress had been made to overcome seeming weaknesses of existing synthetic VHH approaches.

The manuscript needs to be re-worked to highlight the major (and undoubted) advances being made at the “computational” front, which are in fact very interesting.

The manuscript text itself is well written and structured. The figures are clear and mostly self-explanatory. The excel tables provided as supplement are of high quality and are easy as well as interesting to read.

We thank the Reviewer for their thoughtful feedback and excellent suggestions and appreciation of our computational advances for *in vitro* engineering of synthetic VHH antibodies. We address the comments in the revised manuscript and in the response below. To the Reviewer's key points, in the revised manuscript we:

- Added a second affinity maturation and antibody engineering (**new Fig. 5a-d**) that improved our antibodies and show how our approach fits in the general context of synthetic VHHs.
- Expanded testing of VHH clusters to identify additional binders, further showing the strengths of this approach (**new Supplementary Fig. 9**).
- Added biochemical characterization, showing good biophysical properties of VHHs generated by our method, no adverse effects from CDR cysteines (**new Supplementary Fig. 11**), good separation behavior on size-exclusion chromatography (**new Supplementary Fig. 10**) and high thermal stability (**new Supplementary Fig. 12**).
- Text revisions to better acknowledge progress on synthetic VHH libraries, and clarify our specific contributions, especially computationally.

Major points:

1. The authors claim that their approach to generate the synthesis VHH libraries is novel. This might be true at the technical level (namely the orientation-controlled ligation by end blocking using hairpin oligos). However, it is certainly not true for the library design principle, which in fact is based on a consensus design approach taking into account protein sequences of PDB entries of nanobodies. This was exactly the approach used by McMahon et al (Ref 8 in this paper). The only "novelty" was that the authors did randomize the CDRs with NNB codons and thus did not "bias" the amino acid composition of the CDRs to what is found in natural

nanobodies (as had been done for example in Refs. 8, 11 and 13). The use of the cheaper/less sophisticated NNB primers was unlikely a very good idea (see point 2).

We thank the Reviewer for the comment and regret if our use of the term “novel” was misleading and did not clearly explain where our approach fits in the broader context. As the Reviewer notes, we utilized a previously established approach in the design of our synthetic VHH library, and we have modified the language in our manuscript to clarify this (**Page 2, Line 18; Page 6, Lines 108-113**). We also clarify better now the ways in which our approach differs from prior ones (while building on them).

First, our **design** differs from prior designs in both the **length of CDRs** and the **positions selected for randomization**. Such differences likely arise from differences in the size of natural VHH collection retrieved from databases (93 in McMahon et al versus 298 in this study) and/or in how the VHHs are annotated and analyzed (**Methods**). For example, our analysis showed the percentage of VHHs containing CDR2 with lengths 4, 5, or 6 amino acids (a.a). are 32%, 61%, and 1.7% respectively, we thus chose to use CDR2 with a length of 5 a.a. to recapitulate the most prevalent CDR2 length. In contrast, McMahon et al used an equivalent CDR2 length of 4 a.a., while Moutel et al used an equivalent length of 6 a.a. (**Supplementary Fig. 4**). Due to these differences in the detailed features of libraries, we believe it is important to treat these libraries as distinct versions and provide validation of performance. We clarify these differences in the revised **Results (Page 6, Lines 108-113)** and **Methods (Page 23, Lines 464-474)**.

As the Reviewer also noted, we **used NNB codons to randomize CDR positions**. This is a deliberate decision to enable representation of any possible CDR sequences and reduce the cost of the randomizing oligos, making the technology easily accessible. However, as the Reviewer pointed out, NNB codons may cause higher frequency of “undesired” amino acids and potentially impact the biophysical properties of VHHs. To address these concerns, following the Reviewer’s suggestion in points #2 and #3 below, we have added **new experiments and analyses that showed no significant adverse effects associated with the use of NNB codons (new Supplementary Fig. 10-12)**. We discuss these results in detail in the responses to Major

point #2 and #3 below, and are very grateful to the Reviewer for these suggestions, that helped us address this key point.

Finally, we developed a method to **produce our designed library using cell free reactions**: PCR and ligation. As the Reviewer notes, this method uses a new technique to perform ligation in an orientation-controlled manner (**Fig. 1g, Supplementary Fig. 3**). Orientation-controlled ligation simplifies the library generation process, making our method easy to adopt by other researchers. The final library is in the form of linear DNA with each molecule representing a unique member, and the true diversity size of the library can be accurately measured by quantifying the mass of the library sample.

We clarify these distinctions in the revised **Discussion**, to place our approach in the broader context.

2. As a consequence of the NNB oligos, also undesired amino acids are introduced into the CDRs (in particular Prolines and Cysteines). As can be seen in Table S2, in particular cysteines are rarely found in natural VHHs (and if so, then in specific positions and they are then involved in a second SS-bond next to the conserved SS-bond at the centre of the VHH). In the synthetic library presented here, the frequency of cysteine is around 5 % in every randomized positions. Hence, the probability to have a synthetic VHH with at least one extra cysteine (next to the conserved pair) is around 60 % in case of the shortest CDR3 (in total 18 randomized positions for CDRs 1-3; $p = 1 - 0.95^{18}$) and around 72 % for the longest CDR3 (in total 25 randomized positions for CDRs 1-3; $p = 1 - 0.95^{25}$).

We thank the Reviewer for this important point. As we noted above, this is indeed a key distinction of our approach, with some benefits (reduced cost, increased accessibility, broader representation of possible CDR sequences), but some potential drawbacks (if “undesired” amino acids impact the biochemical properties of the VHHs). We were motivated to pursue this path partly because, according to our analysis of natural VHHs, all 20 amino acids are present in the amino acid profile of high diversity CDR positions (e.g., CDR3 positions 7-12, **Supplementary Table 2**). The “desirability” of an amino acid may vary depending on the position in the CDR

and additional sequence context. Indeed, we observed the selection output amino acid profile in different CDR positions shifts away to different distances from the input amino acid profile that are the same across CDR positions (**Fig. 3**), implying that different CDR positions “favor” different amino acid profiles and that it may not be possible to define universally “undesired” amino acids.

The abundance of proline is 5.8% in natural CDR3 (positions 7-12) and 4.0% in NNB randomized amino acid profile (**Supplementary Table 2**), these values are not drastically different with the NNB value slightly lower than the natural value, suggesting that NNB randomization is not a concern regarding proline. The abundance of cysteine is 2.1% in natural CDR3 (positions 7-12), 5.8% in NNB randomized amino acid profile, and 6.0% in the output binders (**Supplementary Table 2**). Although the abundance of cysteines in the NNB profile is more than twice as high as that in the natural profile, cysteine abundance did not decrease in the output profile, implying that an elevated frequency of cysteines did not adversely affect binder selection. In addition, as the Reviewer pointed out in Major point #3, several VHHs containing cysteine in their CDRs are strong binders and SR6 is successfully affinity-matured to become a potent binder and neutralizer, implying that cysteines in CDR do not necessarily affect the function of a VHH.

We clarified this in the revised **Results and Discussion**.

3. Of course one might argue that one does not have to care so much about these cysteines (a quick look at the RBD nanobodies presented in the paper showed that extra cysteines are found in SR1, SR4, SR6 and SR8) and also not about prolines in the CDRs, because i) they were not found to be depleted as part of the selection (see Figure S6) and the respective VHHs containing the extra-cysteines did exhibit binding and viral neutralization. However, the authors have to show non-reduced SDS-PAGE gels of the purified VHHs with binding activity against the RBD, size exclusion chromatography profiles of these VHHs, quantitative affinity measurements (BLI, SPR or similar) and finally some thermal stability data (e.g. via thermofluor using Sypro orange). This reviewer has major doubts whether the identified VHHs are monomers (because of the extra cysteines), are stable and well-behaved on size exclusion (due to “over-randomization”).

Further, the reviewer doubts whether meaningful SPR/BLI data may be recorded (again due to stickiness issues). Synthetic binders have a bad reputation because of poor biophysical behaviour. Hence, a proper biophysical analysis of synthetic binders is really very important, but is completely lacking here.

We are very grateful to the Reviewer for these critical suggestions that we have followed in the revised manuscript. We fully agree with the Reviewer that biophysical characterization of VHHs produced by CeVICA could provide important support for the validation of the system. We therefore performed the following experiments to demonstrate the biophysical properties of VHHs engineered by our system.

First, we performed **non-reducing SDS-PAGE gel analysis of VHHs (Supplementary Fig. 11, Page 14, Lines 291-305)**. Among 7 VHHs containing cysteine in CDRs, 3 showed only a monomer band while 4 showed detectable level of dimer formation after long term storage. However, freshly purified VHH samples did not contain dimers, indicating that dimerization through disulfide bond formation is a slow process. Furthermore, cysteine mediated dimerization of SR6c3 did not have adverse effects on its function as assayed by both ELISA and a pseudovirus neutralization assay. These data indicate that CDR cysteines do not always participate in disulfide bond formation and when CDR cysteines do lead to disulfide formation, the function of the VHH may not be adversely affected. However, it is possible that CDR cysteines may have detrimental effects on some VHHs and the likelihood of which could be reduced by lowering the cysteine abundance of randomizing codons. This can be achieved within our library design and generation method using randomizing DNA oligos synthesized with defined mixture of bases (a service offered by commercial DNA oligo providers such as IDT). For example, when using base mix ratios (A:0.35, T:0.05, G:0.30, C:0.30)(A:0.40, T:0.25, G:0.15, C:0.20)(A:0.05, T:0.05, G:0.45, C:0.45), the cysteine abundance of the randomizing codon can be reduced to 0.37%, while still allowing encoding of all amino acids. This convenient strategy offers fine control over the amino acid profile of the randomizing codon without the need for trinucleotide synthesis. We discuss this possibility in the revised manuscript (**Pages 17-18, Lines 367-369**).

Second, we performed **size-exclusion chromatography of VHHs** and found that the percentage of monomers for the tested VHHs ranges from 90.6% to 98.4%, demonstrating their well-behaved properties in terms of size-exclusion chromatography (**Supplementary Fig. 10, Page 14, Lines 286-287**).

Third, we performed **biolayer interferometry analysis of SR6v15, a new high affinity variant**, and recorded response traces typical of high affinity binders. We showed that SR6v15 has a K_D of 2.18 nM, K_a of $8.79 \times 10^5 \text{ M}^{-1} \text{ s}^{-1}$ and K_d of $1.75 \times 10^{-3} \text{ s}^{-1}$ (**Fig. 5b; Page 13, Line 273**). These values are comparable to those of high affinity VHHs reported by other studies^{1,2}.

Finally, we performed **thermal denaturation and refolding assays to investigate the thermal stability of our VHHs**. Sypro orange dye-based protein thermal shift assay revealed that SR6c3 and SR6v15 both have a T_m of 72°C, consistent with typical values reported for other VHHs³ (**Supplementary Fig. 12a**). We further investigated VHH refolding after heat denaturation at 98°C by comparing **ELISA assay value before and after heating**. We found good refolding for SR6v15 (heated/not heated ELISA absorbance ratio of 0.72, higher than VHH72⁴, 0.33, and Nb21³, 0.57), while SR6c3 has a near 100% refolding (**Supplementary Fig. 12b**). These results show that VHHs engineered by our system have comparable thermal stability compared to VHHs originated from animals³ (**Page 15, Lines 306-319**).

We are very grateful to the Reviewer for this suggestion which helped us improve our study and clarify the capabilities of our approach.

4. An important aspect of the pipeline is the NGS analysis of input and output VHH libraries. In this context, there are two technical aspects which are not touched on in the manuscript. The first one is sequencing errors. Although Illumina is pretty good when it comes to errors, these nevertheless exist. In how far did sequencing errors influence the outcome of the computational pipeline? The second problem is probably more severe. If one uses highly homologous sequences (as is the case here), PCR amplification leads to loop-shuffling (see also Fig. S3 in this paper: <https://www.nature.com/articles/s41592-019-0389-8>). The authors amplified the output/input libraries twice with PCR to introduce the Illumina sequencing adaptors. Consequently, the

composition of the CDRs 1-3 of the sequences obtained are only partially the compositions that were really present in the output/input library. The authors should comment on this problem and may even tweak their computational pipeline to account for PCR-induced shuffling.

We thank the Reviewer for these important questions.

We designed our **NGS analysis pipeline to minimize the effects of sequencing errors and errors introduced during NGS library preparation**, such that low frequency errors in our data will unlikely change the analysis outcome. First, we use a sequence clustering-based approach to select one sequence to represent each cluster (**Methods, *CDR-directed clustering analysis*, Pages 30-31, Lines 639-643**). This representative sequence has the highest cumulative similarity score to all other sequences in the same cluster, such that a sequence containing mutations due to sequencing errors is unlikely to meet this criterion, because it should have positions that are different than most other sequences in the cluster and thus a lower cumulative similarity score. Second, when selecting for beneficial mutations in affinity maturation, the selection decision is based on amino acid frequency change pre- and post-affinity maturation rounds across the entire sequenced population (**Methods, *Identification and ranking of beneficial mutations*, Page 35, Lines 733-753**). Pre and post libraries are always prepared in parallel and sequenced in the same sequencing run, such that systematic sequencing error should be the same for both libraries, and sequencing errors in pre- and post-libraries should cancel each other when calculating the change value. Given the high percentage of experimentally validated binders (78.9%) among the 38 tested VHH clusters and the high efficacy of affinity maturation improvements of SR6, the impact of sequencing error on our NGS analysis pipeline is low. We clarify this point in the corresponding **Methods**.

Regarding **PCR loop-shuffling**⁵, the Reviewer is correct and we have indeed observed it when we were developing and optimizing our system. Briefly, high level of sequence shuffling during PCR will cause selection failure, because a functional VHH will require the right set of CDR1, 2 and 3, when CDRs from different VHH are shuffled together, the resulting “mosaic” VHH will often lose its original binding property. Indeed, in earlier unfruitful attempts of binder selection, we found a CDR from one cluster showing up in many other seemingly unrelated clusters as well

as singleton sequences (sequences not forming clusters), suggesting that CDRs have been shuffled between sequences. We were able to **largely suppress such sequence shuffling during PCR** by optimizing PCR condition to make sure each PCR cycle produces full-length DNA products, notably, by using an enzyme mixture containing DNA polymerases with distinct strengths and sensitivities, and using PCR cycling conditions that are optimized for the primers used in the reaction. We clarify this in the revised manuscript (**Page 28, Lines 574-579, Supplementary Table 3**). This optimized protocol gave a robust increase in RNA yield after three rounds of selection indicating successful selection (**Supplementary Fig. 5a**), and NGS analysis of the output library did not show signs of significant CDR shuffling. The individual cluster files show that the member sequences within each cluster either only contain one set of CDR1+2+3 combination (SR2) or a limited few (SR1, SR4, SR6, SR8, SR12, note that some VHHs' binding property may be retained across a few different CDR3 sequences) (**Supplementary Data 1**). Hence, we believe that sequence shuffling during PCR in our system is controlled to a level that is sufficiently low, such that it does not adversely impact our selection and analysis outcome.

Minor points:

1. Line 55: ref 11 did not produce inconsistent results, as the paper shows that for soluble proteins such as maltose binding protein, one can get really good and biophysically well-behaved binders with ribosome display alone. That paper makes the point that for challenging membrane proteins, change of display systems (ribosome display and phage display) are needed to enrich for specific VHHs. The target protein used in this study (EGFP and RBD) are in fact highly stable soluble proteins, which can be considered as easy targets.

We thank the Reviewer for pointing this out and apologize for the inaccuracy. We have modified the language to better align with the finding of the reference (**Page 3, Lines 52-54**).

2. Line 76: Author statement: "Clustering following high throughput sequencing identifies them more efficiently than methods that rely on the analysis of individual colonies or sequences".

The authors do not provide experimental evidence for this statement.

We have changed the statement to be more in line with the work we did to demonstrate the capabilities of our system (**Page 5, Line 82**). As we did not have data on side by side comparisons, we have removed the statement regarding the comparisons.

3. Line 77: Author statement: “promising a more comprehensive view of the landscape of binder potential, with minimal time and resources.”

I agree with the comprehensive view, but I doubt with the time and resources statement. NGS takes time and is costly. Gene synthesis of VHHs gets cheaper, but is still more costly than getting the clones from minipreps after ELISA analysis. And certainly it takes more time.

We thank the Reviewer for the comment. As we did not perform side by side comparisons and do not have the full information to address costs and human effort for each approach, we have removed the statement regarding time and resources (**Page 5, Line 82**).

4. Line 104: the authors obviously had problems with the oligo qualities, as there seemed to be many indels leading to frameshifts/early stop codons. This is why they used this trick of displaying the VHHs after randomizing CDR1 and CDR2 on the ribosome and capture via a C-terminal Myc-tag. Did the authors use PAGE-purified oligos to assemble the library? If not, this might explain why the problem was so severe. And can the authors give an explanation why the percentage of in-frame sequences did not increase more (only from 25 % to 52 %). Why were the 48 % out-of-frame VHHs (which thus lack the myc-tag) captured/co-enriched as well? My suspicion: they were simply sticking to the anti-Myc antibody or the beads due to poor biophysical properties.

We thank the Reviewer for these questions. The DNA oligos we used for building the input library were synthesized using a standard oligo synthesis service from IDT (www.idtdna.com). The oligos were not PAGE purified, because when mixed bases were used, the actual sequence of individual oligos will have an impact on oligo migration on PAGE gels, thus preventing precise PAGE gel extraction. IDT does not offer PAGE purification of oligos containing mixed bases for this reason. Although in-house PAGE purification of oligos could be performed, we

chose not to do it due to the added complexity to the protocol and that the benefits are not essential. We note this in the revised **Methods** for clarity (**Page 23, Lines 478-481**).

Several factors can in principle contribute to the presence of out-of-frame VHHs after anti-Myc selection: (1) Non-specific binding of RNA or protein to magnetic beads; (2) translation through alternative start codons downstream of areas containing out-of-frame errors; and/or (3) inefficient binding of anti-Myc antibody to the expressed myc peptide that is located between the VHH protein and ribosome. We disfavor (1), because although our *input* library contained 27.5% full-length sequences, the remaining sequences that contained errors do not interfere with full-length sequences and are reduced to <10% after three rounds of RBD selection (**Fig. 2c**), suggesting that these erroneous sequences or their encoded peptides do not non-specifically stick to beads at significant levels to impact binder selection. We note these points in the revised **Methods** (**Page 28, Lines 584-594**).

5. Line 136: Author statement: “The shared clusters likely target the shared components (protein G, anti-Flag antibody) present on the solid support surfaces, and thus represent background binders.”

The authors do not provide experimental evidence supporting this statement.

We thank the Reviewer for raising this point. We have modified the statement to align with the data we provided (**Page 8, Lines 154-155**). We do not validate these background binders because they are not pertinent to the central ideas of the manuscript. It is not certain what their true targets are, and their binding may be too weak to allow us to definitively determine a target.

6. Line 140: top-ranking in terms of what? Match scores of CDR1-3 summed up?

We apologize for the omission. The list is ranked by cluster size. We have added this information in the text (**Page 8, Line 157**).

7. Line 159 and 165: “fit profile” and “fitness cost” not influenced by complete randomization: This seems to be indeed the case in terms of enrichment against the target proteins during the

selection. But as outlined in the major points above, such a “reckless randomization” approach likely compromises the biophysical properties of the VHH proteins.

We thank the Reviewer for this question. Following the Reviewer’s excellent suggestion (Major point #3) we have provided supporting data showing the good biophysical properties of the VHHs we generated.

8. Line 177: “fitness drawbacks in vivo”: this reviewer thinks that there are also major “fitness drawbacks” in vitro, namely in terms of biophysical properties etc. (see comments above).

As noted above, we agree with the Reviewer that biophysical characterization was lacking and have performed experiments to characterize the *in vitro* fitness of the VHHs (See response to Major point #3).

9. Line 209: sequences instead of sequenced.

Thank you! This is now corrected in the manuscript (**Page 12, Line 234**).

10. Line 226: This paper did not optimize the in vitro selection itself.

We have introduced several modifications in our protocol for *in vitro* selection as described in **Methods, In vitro selection**, notably, optimization of PCR condition during recovery stage of the selection cycle to suppress sequence shuffling (**Page 28, Lines 574-579**), as we described in our response to major point #4 and better clarified in the revised **Methods**.

11. Line 339: the materials and methods lacks any downstream processing of the Illumina sequencing data. How did the authors account for sequencing errors? On what basis were sequences discarded (e.g. frameshifts, N reads, etc)?

We regret this omission. We have added further details to the **Methods** to describe processing of Illumina sequencing reads (**Page 26, Lines 533-535**). Briefly, raw reads were separated by index,

trimmed to remove N bases and bases with a quality score of less than 10 prior to downstream analysis.

12. Line 341 ff: the authors did not make any control experiment to show how efficient their ribosome display setup is. A good option is to display a singly high affinity nanobody on ribosomes, followed by a pull-down and quantification by qPCR.

We thank the Reviewer for this great suggestion, which we followed in the revised manuscript. We performed **a control experiment by ribosome displaying SR6c3 then binding to RBD coated magnetic beads then quantification of total bond RNA**. We found that using 100 ng SR6c3 DNA as input yielded 7,910 ng recovered total RNA, among which 989 ng is estimated to be SR6c3 RNA (1/8 of total, calculated by mass ratio of VHH RNA, 649 nt, to *E. coli*. ribosomal RNAs, 4568 nt), representing a coverage rate of 19X in the output. Note that we used direct RNA quantification to measure the yield instead of qPCR because we found that direct RNA quantification was consistent and accurate and can be validated by agarose gel analysis (for example, **Supplementary Fig. 5b**). This control experiment is now described on **Page 29, Lines 596-602**.

13. Line 358 ff: the author did not state volumes and amount of beads used.

We regret this omission. We have added details regarding the magnetic beads used (**Page 27, Lines 560-562**).

14. Line 370: what primer was used to perform reverse transcription?

The sequence of the primer used for reverse transcription are in **Supplementary Table 3, row 64**. We refer to this more clearly from the text (**Page 27, Line 573**).

15. Figure S6c, CDR1 position 1: there must be an error. The sum of the % of the 20 amino acids needs to be 100 % for both axes. But the points are too much shifted to the top/left. Hence the % sum for both axes cannot be the same, although they should.

We regret the lack of clarity in this presentation. Because the natural profile has residues with high abundance (and thus “outlier” data points), such as in CDR1 position 1 R (33.2, 6.5), we set an axes limit of 15 (so that other points can show as well), while providing all numerical values in **Supplementary Table 2**. We note this more clearly in the legend of **Fig. 3 (Page 55, Lines 969-970)** to explain the situation.

Response to Reviewer 2

The authors describe a ribosomal display approach to select VHHs from a semi-synthetic library combined with a bioinformatic approach to identify binders. The overall strategy is very good also the preselection step on myc to increase the percentage of full length VHH binders during library construction is convincing. The affinity matured antibody SR6_c3 shows an inhibition (IC50) of SARS-CoV-2 in the same range as the VH domain antibody ab8 from a phage display library (Li et al 2020). The paper also shows the need for affinity maturation of the VHH binders against RBD using this approach.

We thank the Reviewer for their appreciation of our method and insightful suggestions. We address the Reviewer’s comments and suggestions below and in the revised manuscript.

Major Revisions:

1. line 35: "However, broad application of such in vitro methods remains a challenge...". I can not agree with this statement, because currently 13 FDA/EMA approved antibodies were generated by the in vitro technology antibody phage display (Alfelah et al 2020). These in vitro approaches were also used to generate neutralizing SARS-CoV-2 antibodies (Bertoglio et al 2020, Li et al 2020).

We thank the Reviewer for raising this point and apologize for the inaccuracy. We modified the statement to address the point the Reviewer raised regarding current applications of *in vitro* methods for generating antibodies (**Page 3, Line 35**).

2. the authors should add a negative control antibody in the neutralization assays.

We thank the Reviewer for the great suggestion. We added a negative control VHH in the pseudovirus neutralization assay, and it showed no significant neutralization and matched the result for buffer control (**Fig. 5d**).

3. I suggest to measure the affinities which would show the effect of affinity maturation in addition to the given inhibition assays.

We thank the Reviewer for this important suggestion. Following the Reviewer's comment. We have added a new experiment where we performed affinity measurements **using biolayer interferometry (Fig. 5b) of SR6v15, a new high affinity variant**, recording response traces typical of high affinity binders. We showed that SR6v15 has K_D of 2.18 nM, K_a of $8.79 \times 10^5 \text{ M}^{-1} \text{ s}^{-1}$ and K_d of $1.75 \times 10^{-3} \text{ s}^{-1}$ (**Fig. 5b; Page 13, Line 273**). These values are comparable to those of high affinity VHHs reported by other studies^{1,6}.

4. I'm missing a discussion in this article on the advantage/disadvantages of other ribosomal display or in vitro selection procedures in comparison to CeVICA. I'm also missing a discussion on the generated anti-RBD VHHs.

We agree with the Reviewer that we should have better described our study in the context of other procedures. We have added the **Discussion** section where we discuss (1) Advantages of CeVICA in comparison to other systems. (2) Application areas that CeVICA is particularly suited for. (3) CeVICA contributes to the evaluation of randomizing codon fitness and future library design directions. (4) Properties of anti-RBD nanobodies generated by CeVICA in

comparison to previously generated nanobodies *in vitro* or in animals (**Pages 16-18, Lines 322-391**).

Minor Revisions:

1. 45/46: "screen diversity" > "screening diversity", "library diversity"?

Thank you! We updated the language used (**Page 3, Line 48**).

2. 46/47: "...DNA library delivery into cells (typically <1010)". This is not correct. The "big" naive antibody gene libraries have a library of size >1010 (e.g. Hoet et al 2005, Schofield et al 2007, Glanville et al 2009, Lloyd et al 2009, Kügler et al 2015). Or do the authors mean the transformations efficiency of *E.coli* which is normally in the range of 1E8.

We regret the lack of clarity. We indeed meant the transformation efficiency of *E. coli*. or yeast cells. We clarified this in the revised manuscript (**Page 3, Line 49**).

3. Why did the authors use HEK293T ACE2/TMPRSS2 instead of Vero or Caco cells which are the established cell lines in SARS-CoV-2 neutralization assays?

During the development of the pseudovirus neutralization assay, we found that HEK293T ACE2/TMPRSS2 cells are highly susceptible to pseudovirus infection and produced consistent inhibition measurements, while Vero E6 and Caco-2 cells showed lower susceptibility in our GFP detection-based assays. Note that HEK293T ACE cells have been widely used in SARS-CoV-2 spike pseudovirus neutralization assays^{7,8}. We note this in the revised **Methods (Page 33, Lines 708-712)**.

4. Please give an explanation for the "Gini-Index".

We regret the lack of explanation on this point. The Gini index measures the degree of inequality in a population of individuals (for example, used frequently in studies of income inequality). The

index ranges from 0, indicating perfect equality, to 1, where one member in the population has all the resources. We used $1 - \text{Gini index}$ to indicate the level of diversity for the amino acid profile at each VHH position, where a value of 0 indicates no diversity (one amino acid has a frequency of 100%) and a value of 1 indicates highest diversity (all amino acids have the same frequency). We explain this in the revised manuscript (**Page 22, Lines 450-461**).

5. Please give an explanation for the "Match Score".

Match Score is defined as the ungapped sequence alignment score, calculated for each CDR of two VHHs as the sum of BLOSUM62 amino acid pair scores at each aligned position. We described it in **Methods, CDR-directed clustering analysis (Page 29, Lines 610-613)**.

6. why did the authors isolate 52 identical antibodies from the panning against EGF and RBD? Are these antibodies against the Flag-Tag?

We believe these common VHHs are ones that either non-specifically bind to beads or bind to common proteins on the beads (Protein G, anti-Flag IgG, Flag tag). They are excluded from downstream characterizations because they do not specifically bind to either EGFP or RBD. We do not validate these background binders because their binding may be too weak to allow us to definitively determine a target, we have modified the statement to align with the data we provided (**Page 8, Line 154-155**).

Response to Reviewer 3

The authors develop a method termed CeVICA for the isolation of diverse nanobodies from a synthetic library. This method involves three parts. First, a synthetic library was prepared. This library is based on nanobodies deposited in the PDB, but incorporates large diversity through the randomization of all three CDRs with NNB codons. Second, CeVICA incorporates ribosome display and MACS for the in vitro selection of nanobodies with affinity toward a chosen antigen. Finally, sequences were clustered into families by examining sets of two sequences for pairs of

identical residues in their CDRs. CeVICA then allows for analysis of nanobodies from diverse families.

This method is applied to the generation of nanobodies which bind to and neutralize SARS-CoV-2. Three rounds of selection were performed using the receptor binding domain (RBD) as the antigen. This selection strategy led to the identification of several clusters within the output VHH DNA. Fourteen VHHs representing each of the top clusters were analyzed for binding affinity and neutralization activity. Of the examined nanobodies ~71% (10/14) showed some level of affinity toward the RBD in ELISA, and ~43% (6/14) showed some level of neutralization activity. The CDRs of VHHs in the sorting output was also examined, and the composition of these CDRs was found to be more similar to the input library than the natural VHHs contained within the PDB.

The authors then performed affinity maturation on VHHs isolated from this sorting. An error prone library was generated, sorted for three rounds, and common mutations were identified. Individual clones containing selected mutations and combinations of these mutations were analyzed for binding and neutralization activity. Some nanobodies showed both increased affinity and neutralization activity while others showed only an increase in neutralization activity. Finally, the authors also observed that the error prone sorting introduced some residues consistent with humanization of camelid antibodies.

We thank the Reviewer for their interest and appreciation of our work and we address their comments and suggestions in the point-by-point response below and in the revised manuscript.

Overall issues:

1. It does not appear that this method produces nanobodies with properties that rival the best neutralizing nanobodies from other methods. The best inhibition potency is only ~60 nM. No affinity measurements (KD values are reported). The authors compare their neutralization activity to a previously reported nanobody (VHH-72), but this is a nanobody that was discovered for neutralizing SARS-CoV-1 and also neutralizes SARS-CoV-2. A more fair comparison would

be to compare relative to nanobodies such as Ty1 that were developed specifically for SARS-CoV-2 (Hanke, L., Vidakovics Perez, L., Sheward, D.J. et al. An alpaca nanobody neutralizes SARS-CoV-2 by blocking receptor interaction. Nat Commun 11, 4420 (2020). <https://doi.org/10.1038/s41467-020-18174-5>).

We thank the Reviewer for raising these important points that we address through multiple new experiments in the revised manuscript.

To demonstrate the potential of CeVICA and further improve the potency of our nanobody, we performed a **second affinity maturation** of the most potent nanobody SR6c3. We identified additional beneficial mutations that, when combined, further increased both binding and pseudovirus neutralization potency (**Fig. 5**).

We **compared these new variants to previously reported VHHs as noted by the Reviewer**, including VHH72⁴, Ty1¹ and Nb21³. The most potent new variant, SR6v15 has a K_D of 2.18 nM and pseudovirus neutralization IC50 of 3.59 nM, outperforming VHH72 and Ty1. A dimeric form of SR6v15, SR6v15.d, has an increased potency with IC50 of 0.329 nM, which is comparable to the trimeric Nb21 (Nb21.t), which has IC50 of 0.244 nM. These new data show that CeVICA can reliably improve VHH properties to generate highly potent virus neutralizing agents. We show these results in the **new Fig. 5** and on **Page 13, Lines 269-280**.

We thank the Reviewer for these questions which helped us substantially improve our study.

2. It is unclear if the nanobodies produced by this cell free method have the stabilities and solubilities that would make them useful as reagents or therapeutics. No biophysical data is reported for melting temperatures and %monomer by size-exclusion chromatography, which prevents evaluation of how good these nanobodies are relative to previously generated nanobodies, including those generated via immunization.

We thank the Reviewer for the suggestions. Following this suggestion, we performed size-exclusion chromatography (**Supplementary Fig. 10**), thermal denaturation and refolding

experiments and demonstrated the VHHs generated in this study showed similar stability as VHHs that originates from animals (**Supplementary Fig. 12**). In addition, the purified samples of VHH are **highly soluble** and remain clear without any observable precipitation at concentrations above 3 mg/ml for at least 7 months under storage at 4°C.

Specifically, with **size-exclusion chromatography of VHHs** we found that the percentage of monomers for the tested VHHs ranges from 90.6% to 98.4%, demonstrating their well-behaved properties in terms of size-exclusion chromatography (**Supplementary Fig. 10, Page 14, lines 286-287**).

Moreover, we performed **thermal denaturation and refolding assays to investigate the thermal stability of our VHHs**. Sypro orange dye-based protein thermal shift assay revealed that SR6c3 and SR6v15 both have a T_m of 72°C, consistent with typical values reported for other VHHs³ (**Supplementary Fig. 12a**). We further investigated VHH refolding after heat denaturation at 98°C by comparing **ELISA assay value before and after heating**. We found good refolding for SR6v15 (heated/not heated ELISA absorbance ratio of 0.72, higher than VHH72⁴, 0.33, and Nb21³, 0.57), while SR6c3 has a near 100% refolding (**Supplementary Fig. 12b**). These results show that VHHs engineered by our system have comparable thermal stability compared to VHHs originated from animals^{1,3} (**Page 15, Lines 306-319**).

We are very grateful to the Reviewer for this suggestion which helped us improve our study.

Specific issues and questions:

1. *The observation that, at least at several positions, the residues in the output of the sorting more closely correlates with the representation of residues in the randomly diversified input library than the natural representation of residues in nanobodies from the PDB has the potential to expand the designs of synthetic nanobody libraries available for in vitro selection. However, this observation is based on a relatively small number of sequences from the PDB (298). Would these conclusions regarding the shift away from the natural profile still hold true if a larger database, such as AbYsis, was used to look at the distribution of residues in natural VHHs?*

We thank the Reviewer for this great suggestion and agree that using a natural profile calculated from a larger dataset could consolidate our finding that selection output does not shift towards natural profile but instead is more similar to the input profile, and thus might suggest additional possibilities for design of amino acid profiles for the randomization of CDR positions in synthetic VHH libraries by targeting large scale output library profile.

We thus **retrieved all 1,030 unique VHH sequences from abYsis** (www.abysis.org/abysis, date of download: 2021-05-01) and found that this larger data set had highly similar amino acid profile as our PDB298 data set (with r^2 mean of 0.913 for all CDR positions). We then performed comparative analysis using this larger set to represent natural profile and reproduced all the observations made with the PDB298 dataset. These data are now presented in **Supplementary Fig. 7** and discussed on **Page 10, Lines 195-197**, and **Page 17, Lines 365-369**, where we note the Reviewer's excellent point on the intriguing possibilities for future design. We thank the Reviewer for this excellent suggestion, which helped us improve our study.

2. This paper provides a method for identifying multiple nanobodies directed toward a single antigen with diverse CDRs. It would be helpful to expand upon the advantages of enabling the selection of diverse sequences. For instance, the authors could examine whether this method has the potential to direct antibody binding toward different epitopes within the same antigen; this ability would be particularly useful for antigens in which antibodies are commonly directed toward immunodominant epitopes. Could the authors clarify the need that they aim to address through the selection of diverse nanobody sequences?

We thank the Reviewer for this excellent question. Two features of CeVICA contribute to identification of diverse binders. **First**, the library design and generation method maximize the number of possible CDR sequences it could encode by using non-restrictive NNB codons to randomize up to 25 amino acid positions and by constructing the library with a robust process that is easy to monitor quantitatively. **Second**, the use of cell-free display technology increased the library size that can be screened routinely.

The availability of a diverse set of binders could help identify nanobodies with unique properties. One example is nanobodies targeting rarely targeted epitopes as Reviewer 3 suggested, where output binders are expected to target all the available epitopes of the antigen in a relatively uniform way because of mutual competition between nanobodies targeting the same epitope. Other examples include nanobodies that neutralize viral entry into cells (exemplified by the SARS-CoV-2 neutralizing nanobodies) and nanobodies that can modulate target molecule function (such as receptor agonizing or antagonizing). We have added these points to the revised **Discussion (Page 16, Lines 341-344)**.

3. *“we chose one representative VHH gene from each of the 14 top-ranking RBD unique clusters and validated it for spike RBD binding and SARS-CoV-2 pseudovirus neutralization”*

a. This is an interesting approach for isolating nanobodies with diverse sequences and binding properties. Would other nanobodies contained in the same cluster have similar binding and neutralization characteristics to one another? Specifically, if a nanobody is identified as having good affinity or neutralization, could other nanobodies in this cluster be examined for comparable or improved characteristics?

We thank the Reviewer for this question. We find that all member sequences in the same cluster have highly similar sequences and the frequency of variable residues is insufficient to identify beneficial mutations in a comprehensive and confident manner, especially given the expected frequency of sequencing errors (**Supplementary Data 1**). This is especially true for small clusters, such as SR38 that has only 5 member sequences. To comprehensively improve a VHH, we find the random mutagenesis-based affinity maturation is highly effective.

4. *“In three of the four VHH hallmark residues there were VHHs where the residues were converted to the corresponding human residue as a result of affinity maturation (Fig. S7, arrows). These data imply that at least some of the VHH hallmark residues can be converted to human residues without loss of binding fitness.”*

a. Based on Fig. S7, it seems that, at most, two humanizing mutations occurred in the same sequence and there was a greater increase in VHH hallmark residues at more of the examined

positions. Were individual clones with two or more such mutations (camelid to human) examined for affinity and stability in the absence of a light chain?

We thank the Reviewer for this question. We have examined the effect of incorporating one humanizing residue for SR6, with SR6v10 which contain the R45L mutation. SR6v10 showed similar solubility, binding and pseudovirus neutralization properties as the variant that doesn't contain R45L (**Supplementary Table 8**). It is worth noting that the human IGHV3-23 gene has been successfully used as template for building single domain antibodies⁹ that function independent of light chains, suggesting that the 4 hallmark residues may not present a significant challenge to the biophysical properties of VHHs (**Page 12, Lines 246-249**).

5a. “VHHs were separated into CDRs and frames (segments) by finding regions of continuous sequence in each VHH that best matched to the following standard frame sequences:”

a. How does this methodology align with the CDR and framework definition of more standard numbering systems like Kabat or Chothia? In Table S1, some sequences have only one or two residues in CDR3 (e.g 6SSI, 58HD, 5L21), which seems to suggest part of the CDR may be included in the framework regions using this method.

Following the Reviewer's suggestion, we generated Kabat and Chothia frame annotations of representative VHHs using the abYsis annotate tool (www.abysis.org/abysis/sequence_input/key_annotation/key_annotation.cgi) and compared the annotation with frame annotation generated with our method. All three methods (Kabat, Chothia and ours) showed frame regions with the same core sequence, and with 1-2 amino acid differences in the exact CDR boundaries between the three methods. We chose our CDR boundaries based on the combined frequency of the top two most abundant amino acids, and defined the boundary between frame and CDR where this value drops sharply. The performance of our library suggests our annotation faithfully captured the domain structure of VHHs. We note this in the revised **Methods (Page 22, Lines 443-448)**.

5b. Additionally, could the authors clarify what is meant by “bad sequence” for several of the PDB entries in the sheet labeled “all_VHH_RCSB”?

The entries with “bad sequence” are sequences that are not VHH or are VHH with incomplete sequences. For example, some PDB files contain a VHH bound to their target protein, such that an automatic download retrieves both the VHH sequence and target protein sequence. These non VHH sequences were removed before downstream analysis. The all_VHH_RCSB has been removed because it contains irrelevant information. The 298 VHH sequences used for natural amino acid profile analysis are shown in the unique_VHH_PDB tab in the updated **Supplementary Table 1**.

6. *“We introduced 7 random amino acids for CDR1, 5 for CDR2, and 6, 9, 10 or 13 for CDR3 to match the most commonly observed CDR lengths in natural VHHs.”*

a. In Figure S1, the diversity index in plots C and D show the diversity for 6 positions in CDR2. This seems to disagree with the design of 5 residues in CDR2 for the synthetic VHHs as well plot B and Table S2 which show this position as entirely blank. In Figure S5, the sixth position is again shown as entirely blank in plot A, but no diversity is listed in the diversity index in plot B. Could the authors comment on the change in number and represented diversity for CDR2?

We regret the lack of clarity on this point. We included a sixth position holder in CDR2 when analyzing amino acid profiles to demonstrate that most VHH only has 5 a.a. in CDR2 and the sixth position is mostly empty (only 1.7% of VHH has a CDR2 with 6 a.a., **Supplementary Table 2**). This motivated our choice of 5 CDR2 positions in our library design. The diversity index is always calculated for 6 positions regardless of how many VHH have a sixth CDR2 position, and the index will be 0 if no VHH has CDR2 with the sixth position. Both the natural VHH collection and our input library contained a very small percentage of VHHs having CDR2 with 6 a.a., while the output binder collection has no VHH having CDR2 with 6 a.a., hence the diversity index has a value of 0 for the output binder plot in **Supplementary Fig. 6b** but a non-zero value for natural VHHs and input library in **Supplementary Fig. 1c,d**. We note this in the revised **Method (Page 22, Lines 455-461)**.

7. *“Moreover, correlation of amino acid profiles between output binders and natural VHHs are significantly less than between output binders and input library at most CDR positions (Fig. S6).”*

a. In Fig. S6, are the differences in r^2 values statistically significant at any position? It seems that in CDR2, the output vs natural r^2 values are higher than the input vs output values for at least two to three positions, and in CDR3, the input vs natural values are higher than the input vs output values for at least two positions.

We thank the Reviewer for this important question. The differences in r^2 and RMSE values between output vs. input and output vs. natural are significant for most positions (t test). This information is now added to **Fig. 3**. Note, that RMSE should be used to measure how similar two profiles are, as r^2 does not test the difference directly. In addition, the amino acid profile data is not a normal distribution due to natural profiles strongly favoring one amino acid at certain positions, for example, the natural profile at CDR1 position 1 have R accounting for 33.2%. Because of this, we now use Spearman correlation coefficient to replace our r^2 analyses and have updated the plots in **Fig. 3a** and text in **Page 9, Line 183**.

References

1. Hanke, L. *et al.* An alpaca nanobody neutralizes SARS-CoV-2 by blocking receptor interaction. *Nat. Commun.* **11**, 1–9 (2020).
2. Schoof, M. *et al.* An ultrapotent synthetic nanobody neutralizes SARS-CoV-2 by stabilizing inactive Spike. *Science* **1479**, eabe3255 (2020).
3. Xiang, Y. *et al.* Versatile and multivalent nanobodies efficiently neutralize SARS-CoV-2. *Science* **1484**, eabe4747 (2020).
4. Wrapp, D. *et al.* Structural Basis for Potent Neutralization of Betacoronaviruses by Single-Domain Camelid Antibodies. *Cell* 1004–1015 (2020).
doi:10.1016/j.cell.2020.04.031
5. Egloff, P. *et al.* Engineered peptide barcodes for in-depth analyses of binding protein libraries. *Nat. Methods* **16**, 421–428 (2019).
6. Schoof, M. *et al.* An ultrapotent synthetic nanobody neutralizes SARS-CoV-2 by stabilizing inactive Spike. *Science* **370**, 1473–1479 (2021).
7. Schmidt, F. *et al.* Measuring SARS-CoV-2 neutralizing antibody activity using pseudotyped and chimeric viruses. *J. Exp. Med.* **217**, (2020).

8. Robbiani, D. F. *et al.* Convergent antibody responses to SARS-CoV-2 in convalescent individuals. *Nature* **584**, 437–442 (2020).
9. Li, W. *et al.* High Potency of a Bivalent Human VH Domain in SARS-CoV-2 Animal Models. *Cell* **183**, 429-441.e16 (2020).

Reviewers' Comments:

Reviewer #1:

Remarks to the Author:

The authors have taken all my concerns into consideration and have addressed them either by convincing arguments or additional experiments of high quality. The additional information included now at the technical front (much of it in the methods part) are of high value for VHH aficionados.

I do not have any further concerns and think that this paper really deserves to be published in Nature Communications.

Reviewer #2:

Remarks to the Author:

The authors revised the manuscript very well. I have only one remaining point.

Minor revisions:

- line 35: "However, the adoption of such in vitro methods is still more limited than that of animal-dependent antibody generation(ref 4)".

Why is the in vitro method more limited than animal derived antibodies? The author should give examples, e.g. affinity improved antibodies in vivo in comparison to in vitro selection'. The reference is completely wrong in the context of the statement, because they reference a "political" article by Gray et al on the advantages of in vitro technologies for antibody generation to avoid animal experiments.

Reviewer #3:

Remarks to the Author:

The authors have addressed my concerns. Thank you

Response to Reviewers

Reviewer #1 (Remarks to the Author):

The authors have taken all my concerns into consideration and have addressed them either by convincing arguments or additional experiments of high quality. The additional information included now at the technical front (much of it in the methods part) are of high value for VHH aficionados.

I do not have any further concerns and think that this paper really deserves to be published in Nature Communications.

We thank the Reviewer for their appreciation of our work in our revisions and we are grateful for the Reviewer's comments that helped us substantially improve our manuscript.

Reviewer #2 (Remarks to the Author):

The authors revised the manuscript very well. I have only one remaining point.

We thank the Reviewer for their thoughtful feedback and excellent suggestions.

Minor revisions:

- line 35: "However, the adoption of such in vitro methods is still more limited than that of animal-dependent antibody generation(ref 4)".

Why is the in vitro method more limited than animal derived antibodies? The author should give examples, e.g. affinity improved antibodies in vivo in comparison to in vitro selection'. The reference is completely wrong in the context of the statement, because they reference a "political" article by Gray et al on the advantages of in vitro technologies for antibody generation to avoid animal experiments.

We thank the Reviewer for making this important point and we regret using the wrong reference for the context. By noting that adoption is more limited, we were trying to reflect on the current state of adoption. We now use a different reference, a review of therapeutic antibodies by Lu, R. M. *et al.* 2020, that we believe is better suited to support our statement (**Page 3. Line.36**). And we postulate that the reason why *in vitro* generated antibodies remain a smaller percentage of all approved therapeutic antibodies may include limitations of throughput, functional fitness (including lower affinity due to lack of affinity maturation) and *in vivo* tolerance of antibodies (**Page 3. Line 37**).

Reviewer #3 (Remarks to the Author):

The authors have addressed my concerns. Thank you

We thank the Reviewer for their important comments and feedback that helped us substantially improve our manuscript.